# Probabilistic Spatio-temporal Seasonal Sea Ice Presence Forecasting using Sequence-to-Sequence Learning and ERA5 data in the Hudson Bay region

Nazanin Asadi[1], Philippe Lamontagne[2], Matthew King[2,3], Martin Richard[2], and K Andrea Scott[1]

[1]Department of Systems Design Engineering, University of Waterloo, Waterloo, Canada
[2]Ocean, Coastal and River Engineering Research Centre, National Research Council Canada, Ottawa, Canada
[3]Memorial University of Newfoundland, Newfoundland and Labrador, Canada.

**Correspondence:** Nazanin Asadi (n2asadi@uwaterloo.ca)

**Abstract.** Accurate and timely forecasts of sea ice conditions are crucial for safe shipping operations in the Canadian Arctic and other ice-infested waters. Given the recent declining trend of Arctic sea ice extent in past decades, seasonal forecasts are often desired. In this study machine learning (ML) approaches are deployed to provide accurate seasonal forecasts based on ERA5 data as input. This study, unlike previous ML approaches in the sea-ice forecasting domain, provides daily spatial maps
of sea ice presence probability in the study domain for lead times up to 90 days using a novel spatio-temporal forecasting method based on sequence-to-sequence learning. The predictions are further used to predict freeze-up/break-up dates and show their capability to capture these events within a seven day period at specific locations of interest to shipping operators and communities. The model is demonstrated in hindcasting mode to allow evaluation of forecasted predication. However, the design allows the approach to be used as a forecasting tool. The proposed method is capable of predicting sea ice presence
probabilities with skill during the break-up season in comparison to both Climate Normal and sea ice concentration forecasts from a leading subseasonal-to-seasonal forecasting system.

## 1  Introduction

Spatial and temporal forecasts of sea ice concentration (fraction of a given area covered by sea ice) are carried out at various scales to address the requirements of different stakeholders (Guemas et al., 2016). Short-term forecasts (1-7 days) at high spatial
resolution (5-10 km) are important for day-to-day operations and weather forecasting (Carrieres et al., 2017; Dupont et al., 2015), whereas longer term (60-90 day) forecasts are desired by shipping companies and offshore operators in the Arctic for strategic planning (Melia et al., 2016). In this study we are interested in these longer term forecasting methods, which we will refer to as seasonal forecasting. Typical approaches are usually statistical or dynamical in nature. Statistical approaches include multiple linear regression (Drobot et al., 2006) or Bayesian linear regression (Horvath et al., 2020), whereas by dynamical
approaches we are referring to those that use a forecast model solving the prognostic equations governing evolution of the ice cover (Askenov et al., 2017; Sigmond et al., 2016). An excellent overview of both statistical and dynamical approaches is given in Guemas et al. (2016).

An early study on seasonal sea ice forecasting using a dynamical approach is that by Zhang et al. (2008), in which they evaluated the ability of an ensemble of sea ice states from a coupled ice-ocean model to predict the spring and summer Arctic sea ice extent and thickness for the year of 2008, following the anomalously warm year of 2007. Each ensemble member was generated by forcing the coupled ice-ocean model with atmospheric states from one of the previous seven years, and running the model forward in time for one year. Similar to Zhang et al. (2008), the majority of studies on sea ice prediction and forecasting focus on the pan-Arctic domain. A comparison between pan-Arctic and regional forecast skill was carried out by Bushuk et al. (2017), where skill was assessed using the detrended anomaly correlation coefficient (ACC) of sea ice extent. It was shown that the ACC of seasonal forecasts in specific regions was dependent on the region and forecast month.

Dynamical forecast models solve differential equations describing the physics of the underlying system. Solution methods for these types of equations are well-known and relatively robust. A key challenge with these models in operational forecasting is the high level of computational resources required to generate a forecast. This disadvantage can be overcome by using statistical approaches such as multi-linear regression (Drobot et al., 2006) or canonical correlation analysis (Tivy et al., 2011). Both of these approaches determine a linear relationship between a set of predictor variables and a set of predictands.

More recently, convolutional neural networks (CNNs), which are able to learn nonlinear relationships between spatial patterns in input data and predictands, have been used for sea ice concentration prediction (Kim et al., 2020). The study by Kim et al. (2020) used eight predictors composed of sea ice concentration data and variables from reanalyses to train 12 individual monthly models, and produced monthly spatial maps of sea ice concentration (SIC). Their method was able to predict mean September sea ice extent in good agreement with that from from passive microwave data, evaluated for the year of 2017, where the sea ice extent is the total area in a given region that has at least 15% of ice cover. Similar to Kim et al. (2020), Horvath et al. (2020) focused on the September sea ice extent. Hovath et al. (2020) used a Bayesian logistic model to predict both a monthly average sea ice concentration and an uncertainty. The model inputs were atmospheric and oceanic predictor variables and sea ice concentration from satellite data. It was found that the uncertainty was higher at the ice edge, although further analysis of this output was not given. Another recent study, (Fritzner et al., 2020) compared two machine learning (ML) approaches, K-nearest-neighbour (KNN) and fully convolutional neural networks, with ensemble data assimilation.

A recent approach close to the one presented here is IceNet (Andersson et al., 2021), which trained an ensemble of CNNs to produce monthly maps of sea ice presence (probability SIC > 15%) for forecast lengths up to 6 months. Similar to other studies, input to this model consisted mainly of reanalysis data. A novel aspect was the training protocol, which consisted of pre-training each ensemble member using a long time series of data from the Coupled Model Intercomparison Project phase 6 (CMIP6) and then fine-tuning the trained CNNs using sea ice concentration observations, followed by a scaling method, known in the ML community as 'temperature scaling', to produce a calibrated probability of sea ice presence.

None of the previously proposed ML approaches produce a forecast that propagates in space and time, or a *spatio-temporal* forecast. In this study we investigate a sequence-to-sequence (Seq2Seq) learning approach to provide daily spatio-temporal forecasts of the probability of sea ice presence (probability SIC > 15%) over the region of Hudson Bay, with forecast lead times up to 90 days. To keep the method general, we use ERA5 data as input to our model. By using the Seq2Seq approach we are able to produce forecasts over a different number of days than our training sequence. The method is similar to operational

forecasting studies (Chevallier et al., 2013; Sigmond et al., 2013), where an initial state is propagated forward in time, except we are using a data-driven machine learning approach, as compared to a physics-based model, and our forecasted variable is a number between 0 and 1 that indicates an (uncalibrated) probability of sea ice presence at a grid location, as compared to sea ice concentration.

## 2 Data

### 2.1 ERA5

The present study utilizes ERA5 reanalysis data for model predictors and validation (Hersbach et al., 2018). ERA5 is a recent reanalysis produced by the European Center for Medium Range Weather Forecasting (ECMWF). It consists of an atmospheric reanalysis of the global climate providing estimates of a large number of atmospheric, land and oceanic climate variables. The spatial resolution is ≈31 km and reanalysis fields are available every hour from 1979 - present (Wang et al., 2019). Observations are assimilated into the atmospheric model using a 4D-Variational data assimilation scheme. In this study ERA5 reanalysis data from 1985-2017 are used.

The sea ice concentration data used in ERA5 is from the EUMETSAT Ocean and Sea Ice Satellite Applications Facility (OSI-SAF) 401 dataset (Tonboe et al., 2016). These data are produced using a combination of passive microwave sea ice concentration retrieval algorithms to benefit from low sensitivity to atmospheric contamination of the surface signal, while maintaining an ability to adapt to changes in surface conditions through the use of variable tie-points for ice and water (Tonboe et al., 2016). Although the SIC is gridded to a 10 km grid, the spatial resolution of these data is limited by the instrument field of view of the 19 GHz channel used in the SIC retrieval, which is 45 km × 69 km. When the SIC data are ingested into ERA5, the SIC values that are less that 15% are set to zero. Additionally, SIC is set to zero if sea surface temperature (SST) is above a specified threshold to account for known biases in passive microwave sea ice concentration during melt.

The current study utilizes daily samples with the following eight input variables from ERA5 dataset over the period of 1985-2017: sea ice concentration, sea surface temperature, 2m temperature (t2m), surface sensible heat flux, wind 10 meter U-component (u10), wind 10 meter V-component (v10), landmask and additive degree days (ADD) derived from the t2m variable. All the input variables except sea ice concentration and landmask are normalized before being input to the network. Recalling that data are available from ERA5 every hour, the fields from 12:00 (midday) were used.

There were some irregularities with the ERA5 landmask file and the sea ice concentration. Some locations indicated as land in the landmask file had a non-zero sea ice concentration value. At these locations the sea ice concentration was set to zero. There were also some locations indicated as non-land in the landmask file that had a zero ice concentration, even when the ice concentration should be non-zero based on the atmospheric conditions, season, and examination of the time-series at the given location. At these locations we set the sea ice concentration to the average of the non-land neighboring pixels.

## 2.2 Operational Ice Charts

Regional ice charts from the Canadian Ice Service (CIS), referred to herein as CIS charts, were used to complement the use of ERA5 for verification of freeze-up and break-up dates. CIS ice charts are compiled by analysts who manually combine data from various sources, including synthetic aperture radar imagery, sea ice concentration from passive microwave data, optical data and ship reports. Regional ice charts are available on a weekly or biweekly basis over the study period and fully cover the study domain. Although daily ice charts are available at a higher temporal frequency than regional charts, they are only available during certain times of the year and over certain regions. Due to this non-standard spatio-temporal coverage, daily ice charts were not used. It is important to note the temporal resolution of the CIS regional ice charts is coarse compared to the needs of this assessment.

## 2.3 S2S Forecasts

The subseasonal-to-seasonal (S2S) system by ECMWF (Vitart, F. and Robertson, A.W., 2018) was used to complement the use of ERA5 for verification of binary accuracy spatially and across seasons. The S2S predictions are launched twice a week (Monday and Thursday), with forecasts for lead times up to 46 days. For the comparison presented here, the data from our models are extracted for the same launch dates as those used for the S2S system. The S2S data was extracted at a spatial resolution of $0.25^o \times 0.25^o$, and interpolated to our 31 km grid resolution using a nearest neighbour approach. Results are shown only for 2016 and 2017 because these are the years for which forecasts are available for the S2S system that overlap with our study period.

## 3  Study Region

For the present study we focus on the Hudson Bay System, consisting of Hudson Bay, James Bay, Hudson Strait and Foxe Basin (Fig. 1). The area is bordered by 39 communities, where 29 of them are exclusively accessible by sea or air. These communities rely extensively on sealift operations during the ice-free season to receive their yearly resupply of fuel and goods too heavy to be flown. Shipping traffic, mostly confined during the ice-free and shoulder seasons, is also generated by mining, fishing, tourism and research activities (Andrews et al., 2018). The study area is seasonally covered by first-year ice, with open water over most of the domain each summer, with the exception of some small regions in Foxe Basin. The seasonal cycle of ice cover in this region is dominated by local atmospheric and oceanic drivers (Hochheim and Barber, 2014). Freeze-up generally starts in November (earlier in the northern part of the region) and lasts for a couple of months. Break-up usually starts in May or June, and the break-up period is a little longer than freeze-up, at 2-3 months. Recent years show earlier break-up and later freeze-up. The trends and their significance are dependent on the region (Hochheim and Barber, 2014; Andrews et al., 2018).

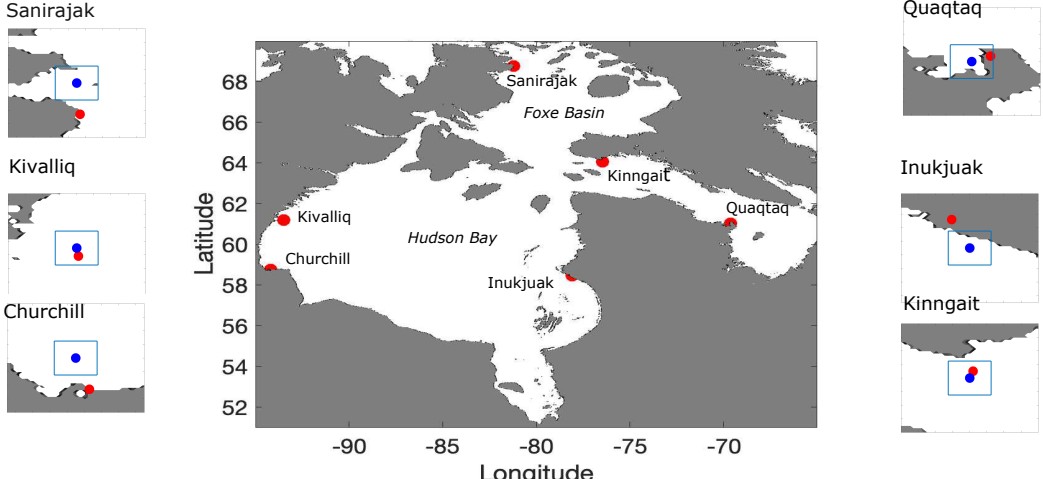

**Figure 1.** The study region with locations of interest shown in red. The insets show the location of a nearby port or polynya (red) and the nearest point on the model grid (blue) that is outside of the land boundary (where landmask from ERA5 is less than 0.6), in addition to a bounding box that approximates a grid cell. The ports are Churchill, Inukjuak and Quaqtaq, whereas the polynyas are near Sanirajak, Kivalliq and Kinngait.

## 4   Forecast model architecture

The seasonal forecasting problem of this study can be formulated as a spatio-temporal sequence forecasting problem that can be solved under the general sequence-to-sequence (Seq2Seq) learning framework (Sutskever et al., 2014). In Seq2Seq learning, which has successful applications in machine translation (Cho et al., 2014), video captioning (Venugopalan et al., 2015), and speech recognition (Chiu et al., 2018), the target is to map a sequence of inputs to a sequence of outputs, where the inputs and outputs can be different lengths. The architecture of these models normally consists of two major components; encoder and decoder. The encoder component transforms a given input (here, a set of geophysical variables such as sea ice concentration, air temperature etc.) to an encoded state of fixed shape, while the decoder takes that encoded state and generates an output sequence (here, a sea ice presence probability) with the desired length, which here is the number of days in the forecast (90 days).

For this study, following the encoder-decoder architecture described above, two spatio-temporal sequence-to-sequence prediction models are developed. These will be referred to herein as the "Basic Model" and the "Augmented Model" and are described in Sections 4.1 and 4.2 respectively. For both models, the prediction sequence is unrolled over a user-specified number of forecast days to produce ice presence probability forecasts on a spatial grid each day with a scale of $\approx 31$ km (the same as the ERA5 input data).

## 4.1 Basic model

The encoder section of the Basic model takes the geophysical variables from the last three days (sea ice concentration, air temperature etc.) as input. Each input sample is of size $(3 \times W \times H \times C)$ where 3 is the number of historical days, $W$ and $H$ are the width and height of the raster samples in their original resolution and $C$ is the total number of input variables (here eight). Using a longer input sample of 5 days was also tested, but did not lead to an improvement in forecast quality. With this longer input the quantity of data to be processed was greater than that for 3 days, which increased the computational expense and data storage requirements, hence 3 days were used for the experiments shown here.

The overall architecture is shown in Fig 2a. The encoder starts by passing each input sample through a feature pyramid network (Lin et al., 2017) to detect spatial patterns in the input data at both the local and large scales. Next, the sequence of feature grids extracted from the feature pyramid network are further processed through a convolutional LSTM (long short-term memory) layer (ConvLSTM) (Hochreiter and Schmidhuber, 1997; Xingjian et al., 2015), returning the last output state. This layer learns a single representation of the time series that also preserves spatial locality. The most recent day of historic input data is concatenated with the ConvLSTM output to better preserve the influence of this state on the model predictions. The encoder provides as output a single raster with the same height and width as the stack of raster data input to the network, but with a higher number of channels such as to fully represent the encoded state. The final encoded state is then fed to a custom recurrent neural network (RNN) decoder that extrapolates the state across the specified number of time-steps. It takes as input the encoded state with multiple channels and as output produces a state with the same height and width as the input over the desired number of time-steps in the forecast (here 90 days). Finally, a time-distributed network-in-network (Lin et al., 2013) structure is employed to apply a 1D convolution on each time-step prediction to keep the spatial grid size the same but reduce the number of channels to one, representing the daily probabilities of sea ice presence over the forecast period (up to 90 days).

The custom RNN decoder, shown in Fig 2b, as is common of many RNN layers, maintains both a cell state and a hidden state (Yu et al., 2019). First, the initial cell state and hidden state are initialized with the input encoded state. Then, at each time-step and for each of the states, the network predicts the difference, or residual, from the previous state to generate the updated states using depthwise separable convolutions (Howard et al., 2017). The output of the decoder section is the concatenation of the cell states from each time-step.

## 4.2 Augmented model

A slight variant of the Basic model, referred to as the Augmented model, is developed to accept a second input. This second input has the same height and width as the first input but corresponds to Climate Normal of three variables over the required period (e.g. 90 days), where these variables are t2m, u10 and v10 and their Climate Normal is calculated from 1985 to the last training year for each forecast day. These variables were chosen because of their availability in both historical data sets, and real time (for this application, through the Meteorological Service of Canada GeoMet platform). Since this branch of the network 'augments' the core model, it was desired to keep this flexibility for future development as our computing infrastructure is designed to connect with GeoMet. For the augmented model, the original encoder structure for historical input data remains

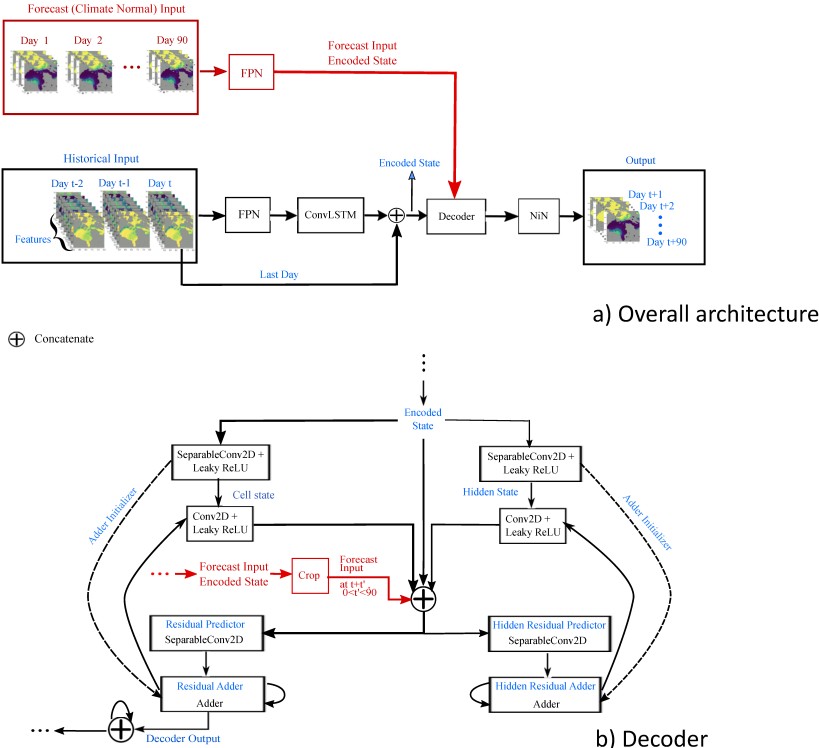

a) Overall architecture

b) Decoder

**Figure 2.** Overall network architecture (a) and custom decoder (b). The black portion in panel (a) refers to the Basic model while the red portion in panel (a) refers to the additional components required for the Augmented model. The dashed arrows show a process carried out only once (the initialization of the adder). FPN refers to the feature pyramid network, ConvLSTM, the convolutional long short-term memory network, NiN is the network in network module.

unchanged, but a secondary encoder is added to the network, consisting of a feature pyramid network, that receives the Climate Normal data as input. A secondary variant of the decoder component is implemented, which accepts this encoded sequence in order to enhance estimates of the residuals at each of the future time-steps (see Fig 2).

## 5    Description of Experiments

Since the overarching goal is to provide a tool to stakeholders that can be used operationally, a training and validation protocol is required that truly assesses the forecasting skill without using future data. For example, on this basis a leave-one-out approach cannot be used. Instead, we initially train over a given number of years, then we update the model weights for future training periods, where the model weights are the learned parameters that transform the input to the output. We tested different initial training periods (10 years vs 20 years) and also different numbers of months to include in training our monthly models. The current protocol (Fig 3) led to the best results. In this protocol for each month of a year a separate model is trained on data from the given month as well as the preceding and following month. For example, the 'April model' is trained using data from

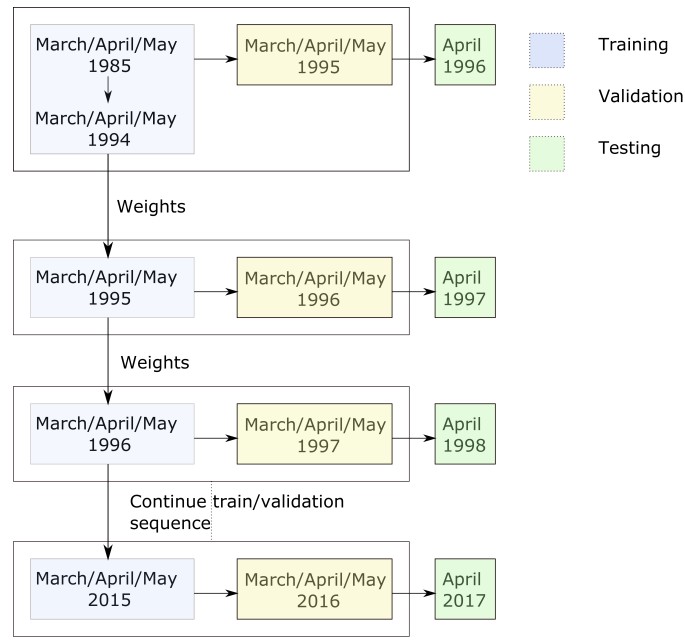

**Figure 3.** Training, validation and test protocol used for both the Basic and Augmented model.

March 1 to May 31. This monthly model is initially trained on data from a fixed number of years, chosen to be 10 years, as a compromise between having enough data to provide the model with representative conditions from which it can learn, while allowing enough data to be set aside for validation and testing. After this initial experiment, to predict each following test year $i$, using a rolling forecast prediction, the weights of the model from year $i-1$ are updated with data from year $i-2$. Data for year $i-1$ is used as validation for early stopping criteria and to evaluate the training performance. For example the 'year 2003' model is initialized with weights from the year 2002 model, which are updated with data from year 2002, validated on year 2003, and used to predict year 2004. This process is used to produce forecasts of sea ice presence for years 1996 to 2017. Since the weight updates use only data of one year for training and validation, the method is computationally fast and efficient.

The ML models are implemented using the TensorFlow Keras open-source library with stochastic gradient descent (SGD) optimizer with learning rate of 0.01, momentum of 0.9 and binary cross-entropy loss function. The maximum training epoch for the initial model and the retraining process is 60 and 40 respectively and for both cases the training process stops if the validation accuracy is not improved after 5 epochs.

## 6 Skill Scores

In order to evaluate the performance of the ML models, the binary accuracy, Brier score and accuracy of freeze-up and break-up dates are used. The main observations to which model forecasts are compared to are the ERA5 sea ice concentration (thresholded at 15% to convert to sea ice presence). Based on our training, verification and testing procedure, the ERA5 states

used as observations are from future dates, hence there is a degree of independence from the data used to train the model. To provide a baseline, we also compare our ML models to a Climate Normal, which is defined here as the average of the ERA5 sea ice presence from 1985 to the last year in the training set for each experiment. While inputs of each model in training and test procedure are derived from three months of each year, only the results from the central month (2nd of three) are selected to evaluate the results of the given model. For example, the April model is trained using historical data from March-April-May. This model is evaluated using 90-day forecasts launched during the month of April. Other data sets used for comparison are sea ice concentration from the ECMWF S2S forecasting system, and operational ice charts (described in Section 2.2).

## 6.1 Binary Accuracy

Binary accuracy is calculated by mapping the ML model forecasts, which denote a probability of sea ice presence, to binary values by thresholding the probability such that when $P > 0.5$ the pixel is considered to be ice and when $P \leq 0.5$ it is considered to be water (similar to Andersson et al. (2021)). After this thresholding, the binary accuracy is calculated as $(TP + TN)/N$, where $TP$ denotes a true positive, and has a value of one if both the pixel in the model and observations are one (indicating ice), and a value of zero otherwise; $TN$ denotes a true negative, and has a value of one if both the pixel in the model and observations are zero (indicating water); and $N$ is the total number of pixels considered. When binary accuracy is used to calculate monthly scores for the entire domain (Fig 4), $N$ is the product of the number of pixels in the spatial domain, the days in the given month and the number of years over which the forecasts are evaluated. For binary accuracy a score of one is considered optimal.

## 6.2 Brier score

Binary accuracy scores do not differentiate between a predicted probability of 0.51 and 0.9. Both would be a true positive if the pixel is ice in the observations. Small changes in the predicted probability around the probability threshold impact the binary accuracy. An alternative score that better reflects the value of the predicted probability is the Brier score (BS) (Ferro, 2007),

$$BS = \frac{1}{M} \frac{1}{T} \sum_{i=1}^{M} \sum_{t=1}^{T} (P_{t,i} - O_{t,i})^2, \tag{1}$$

where $P_{t,i}$ is the model prediction (sea ice presence probability) and $O_{t,i}$ is the corresponding observation (zero or one), at time $t$, and pixel $i$; $M$ represents the total number of pixels in the spatial domain and $T$ the total number of temporal outputs used (note $N = M * T$). For the Brier score a value of zero is considered optimal.

## 6.3 Freeze-up and Break-up Accuracy

The accuracy of the model in predicting freeze-up and break-up dates is indicative of operational capability of the trained models to support shipping operations during the shoulder season. Following the definition used by the Canadian Ice Service (CIS), the freeze-up date of each pixel in a year is the first date in the freeze-up season (October 1st to January 31st for Hudson Bay) that ice (value of 1 after thresholding the predicted probabilities at 50%) is observed for 15 continuous days. A

similar procedure is carried out to predict break-up, with the exception that the pixel must be considered water (value of 0 after thresholding its predicted probability) for 15 continuous days for break-up to have occurred in the break-up season (May 1st to July 31st for Hudson Bay, with forecasts initialized up to July 31st considered). These per pixel per year freeze-up/break-up dates are calculated for observations, Climate Normal and model predictions at 30 and 60 lead days. To obtain each accuracy map, first the predicted and observed freeze-up/break-up dates per pixel per year are compared. If the 2 dates are within 7 days of each other, the prediction is correct (a value of one is assigned), and if not, the prediction is incorrect (a value of zero is assigned). Then, the results are averaged over the total number of years to obtain an overall score between 0 and 1, which we will refer to as freeze-up/break-up accuracy.

## 7    Results

### 7.1    Forecasts of Ice Presence

#### 7.1.1    Monthly averaged results

For each day in the test set, which is the set of days over which the 90 day predictions are launched, we have 90 binary accuracy maps of our study region. The monthly statistics are summarized in Fig 4(a,b,c). The value at index $(i, j)$ of each panel of Fig 4(a,b,c) represents the average binary accuracy score of all predictions in the test set that are launched at month $i$ at lead days $j$ where $1 \leq i \leq 12$ and $1 \leq j \leq 90$. The $(1, 1)$ index value of Fig 4a shows the binary accuracy of 1-day forecasts launched between January 1 and 31, ending January 2 to February 1. The $(1, 2)$ index value corresponds to the binary accuracy of 2-day forecasts. These forecasts were launched between January 1 to 31 ending January 3 to February 2. The $(2, 1)$ index value corresponds to the binary accuracy of 1-day forecasts launched between February 1 and 28, ending February 2 to March 1.

Binary accuracies (Fig 4 a,b,c) are close to 100% for the month of January and for lead times that cover the months of January, February and March, as would be expected, because at this time the region is covered with ice. In contrast, for forecasts at the beginning of the open water season (June and July), Climate Normal struggles to accurately capture the ice cover for lead times of 1 to 50 days (Fig 4a), likely due to inter-annual variability and lengthening of the open water period (Hochheim and Barber, 2014; Andrews et al., 2018). The Basic and Augmented models have higher accuracies than Climate Normal over these months (Fig 4 d,e). We also note improvements in the Basic and Augmented models at short lead times for August, September, October and November, as compared to Climate Normal (Fig 4 d,e). Improvements with the Augmented model can be seen in particular for longer lead times in July/August, and at shorter lead times (15-50 days) for November (Fig 4f). These forecasts correspond to the freeze-up period, which starts in mid-October or November in the study region and lasts for approximately two months (Hochheim and Barber, 2014).

Figure 5 presents the monthly averaged Brier scores for the Basic and Augmented models (Fig 5 a,b), and their differences (Fig 5c) as a function of lead days. Similar to Fig 4, each value at index $(i, j)$ of panel (a) and (b) of Fig 5 represents the average Brier score of all predictions in the test set that are launched at month $i$ at lead days $j$ where $1 \leq i \leq 12$ and $1 \leq j \leq 90$. The resulting pattern is similar to that for binary accuracy. Recalling that a Brier score of zero is optimal, the higher Brier scores seen

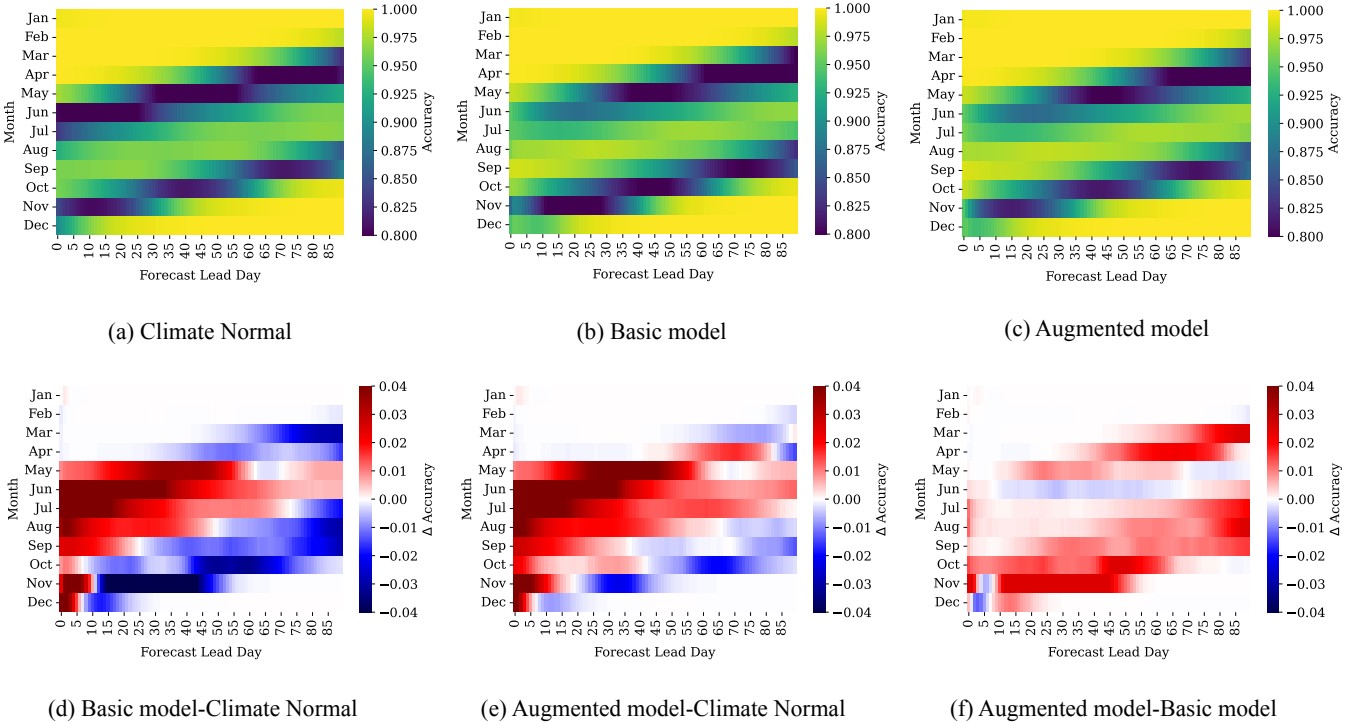

**Figure 4.** Binary accuracies as a function of lead time. Top row panels show the binary accuracy of each model (a)-(c) while bottom row panels show the differences in binary accuracy between the models (d-f). The Augmented model is trained with additional 90-day Climate Normal input. Most differences are observed in the break-up and freeze-up seasons.

during freeze-up and break-up for both models indicates poorer performance during these seasons. Their difference (Fig. 5c) indicates a better score for the Augmented model at longer lead times especially for March, April, July and August. In contrast for some cases like 60-90 lead days forecasts of the September model the Brier score of the Basic model is around 0.01 better than the Augmented model. The reason for the higher Brier score of the Basic model in comparison to the Augmented model may be because the September model uses training data over August-October. The trend over this period may be less representative of more recent ice conditions (Hochheim and Barber, 2014; Andrews et al., 2018), which may make the additional data used in the Augmented model un-helpful at these longer forecast periods.

The calibration curves of the Basic and Augmented September models are shown in Fig 6. These curves represent the observed frequency of ice presence, where the frequency is calculated over the entire domain, versus the forecasted probabilities for different lead days of forecasts launched in September. For short lead times both models, especially the Basic model, show close to perfect calibration (blue line) but at 60 lead days the underestimation is more significant for the Augmented model with lower forecasted probabilities of ice in comparison to observations, while at 90 lead days the overestimation is more significant for the Augmented model, with a forecasted probability that is much higher than the observed frequency of ice. This suggests that in comparison to observations, freeze-up may be delayed at 60 lead days for regions with freeze-up dates

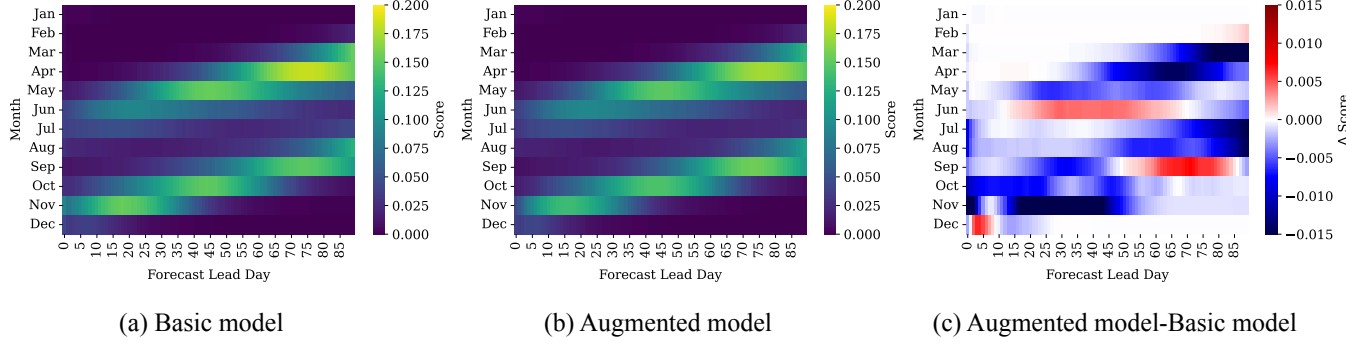

| (a) Basic model | (b) Augmented model | (c) Augmented model-Basic model |

**Figure 5.** Brier score of the Basic (a) and Augmented (b) model as a function of lead time. Their score difference is shown in (c). Most differences are observed in the break-up and freeze-up seasons.

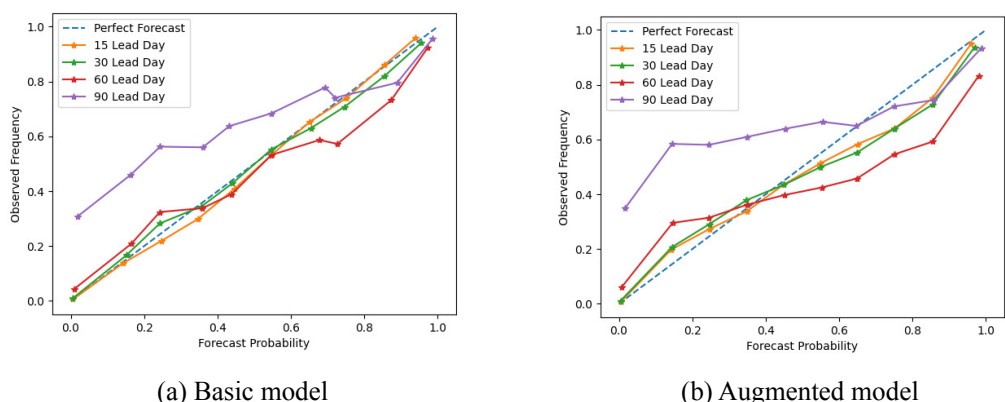

| (a) Basic model | (b) Augmented model |

**Figure 6.** Calibration curves for the Basic and Augmented model for sea ice presence forecasts initiated from September 1st to 30th for different lead days. At 90 lead days, both models underestimate the probability of sea ice when observed frequencies are less than 0.75 and overestimate the probability of sea ice at higher probabilities. The Basic model is well-calibrated for short lead times.

around November and may be too early at 90 lead days for regions with freeze-up date around December for the Augmented model.

### 7.1.2 Spatial maps of sea ice presence

270 Binary accuracy values averaged over the domain and each month do not provide information about model performance at each location in the spatial domain, or at a finer time scale. The model proposed here provides a spatial map of the probability of ice for each day in the forecast period. In Fig 7 the spatial distribution of ice and water is shown with the probability of ice for three different dates during the break-up period. The observations are ice and water obtained by applying a threshold of 15% to SIC from ERA5 for the given date. For example, given that the forecasts are launched on May 6th 2014, the left column 275 (after 30 days) corresponds to the sea ice state on June 5th 2014.

The forecast after 30 days indicates both the Basic and Augmented models predict a reduced ice presence probability along the east coast of Hudson Bay that is in better agreement with observations than Climate Normal. Similarly, after 50 and 70 days, Climate Normal has a higher ice cover relative to the lower probability of ice for the Basic Model in the central part of Hudson Bay, while the Augmented model is in better agreement with observations. In comparing the probability maps for the Basic and Augmented model, it can be noted the Basic model has reduced ice presence probability in the southern part of the domain. Overall we find the spatial pattern of break-up to be in good agreement with the observations, in particular for the Augmented model.

## 7.2 Assessment of Operational Freeze-up and Break-up date forecasting

### 7.2.1 Freeze-up and Break-up in comparison with ERA5 data

We start our comparison using ERA5 as the baseline, consistent with our earlier comparisons. Figure 8 and 9 show the freeze-up and break-up accuracy maps of the Climate Normal as well as the Basic and Augmented model for 30 and 60 lead days. The freeze-up accuracy maps (Fig 8) show similar spatial patterns for the Basic and Augmented models for 60 lead days, with differences between the two models for 30 lead days. The freeze-up accuracy for 30 lead days looks very different also from Climate Normal. To investigate the prediction of freeze-up by the Basic model at 30 lead days, we looked at forecasts from the November and October models for 30 and 60 lead days respectively, as freeze-up mainly happens in December. It was found (not shown) that the December sea ice presence accuracy of the Basic model at 30 lead days is lower in the central region and higher in Hudson Strait compared to other methods, which explains the difference in freeze-up prediction maps. The poorer accuracy in the central region is because freeze-up was too late, discussed further in Section 8.

In contrast to freeze-up, for the break-up accuracy the Climate Normal (Fig 9a) has an overall poor accuracy, while the Augmented model at 30 lead days (Fig 9d) has the best accuracy, especially in the central region. The break-up prediction accuracy degrades at 60 lead days for both the Basic and Augmented models.

The interannual variability of accuracy in freeze-up (October 1st to January 31st) and break-up (May 1st to July 31st) predictions is presented in Fig 10 for 30 and 60 lead days. The respective trends are shown by dashed lines. While no significant trend is observed for freeze-up accuracy at both lead times, the break-up accuracy (Fig 10 c,d) shows a declining trend of 2%. Similar to Fig 9 for freeze-up/break-up date predictions, both the Augmented and Basic models have their highest improvement compared to Climate Normal for break-up at 30 lead days. In addition, for both cases, 2010 shows an extreme case where Climate Normal has the lowest accuracy over the entire period. For that year, the Augmented model has a lower freeze-up accuracy compared to other years, while its break-up accuracy does not show any significant variability over the years. It has been noted in an earlier study that 2010 was an anomalous year (Hochheim and Barber, 2014).

The ability of the model to predict freeze-up and break-up dates can provide helpful information for local communities and shipping operators. Here, the nearest pixels to three sample ports shown in Fig 1, Churchill, Inukjuak, and Quaqtaq, and Sanirajak (formerly known as Hall Beach) are selected. The sites were chosen because they represent locations with significantly different sea ice conditions. Churchill and Inukjuak are located on the east and west coasts of Hudson Bay, with

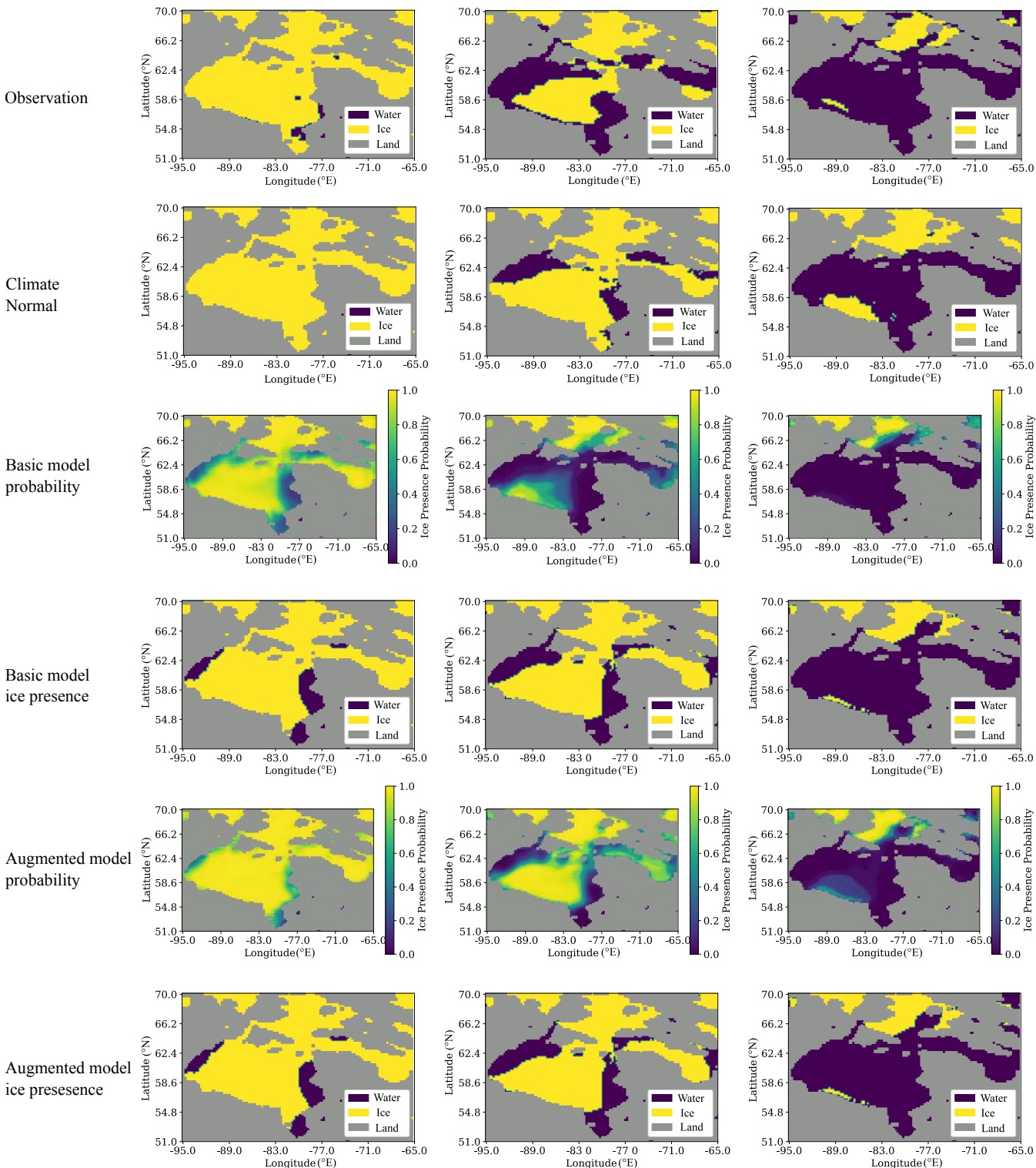

**Figure 7.** Spatial patterns of sea ice presence during break-up. These models are trained using data from May, June, and July. The forecasts are launched on May, 6th, 2014 and are displayed after 30 days (June 5th), 50 days (June 25th) and 70 days (July 15th).

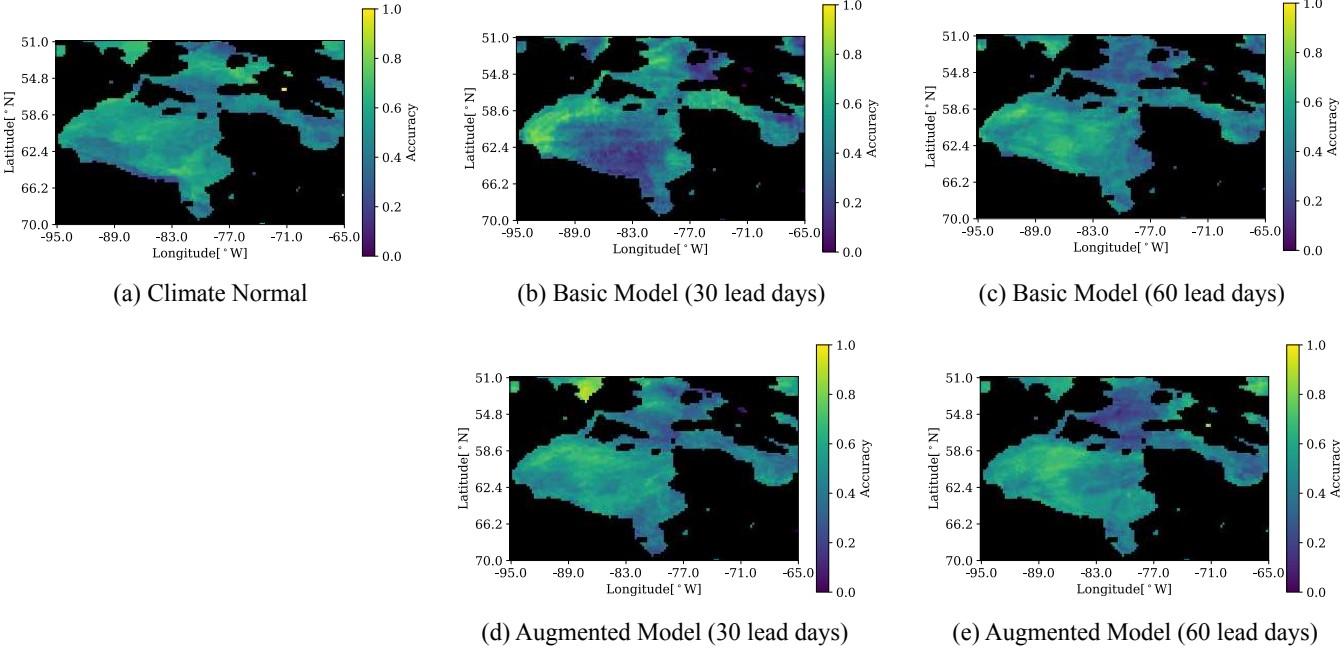

(a) Climate Normal     (b) Basic Model (30 lead days)     (c) Basic Model (60 lead days)

(d) Augmented Model (30 lead days)     (e) Augmented Model (60 lead days)

**Figure 8.** Accuracy of predicted freeze-up date within 7 days. Freeze-up dates are checked from October 1st to January 31st. The 7-day window is chosen to match to definition used by the Canadian Ice Service.

Churchill being a major port as part of the potential Arctic Bridge shipping route. The east coast is significantly impacted by
310 freshwater inflow from rivers draining into Hudson Bay, while the west coast region is impacted by northwesterly winds (there is a latent heat polynya, the Kivalliq polynya, that runs along the northwest shore of Hudson Bay (Bruneau et al., 2021)). There are additionally east-west asymmetries in Hudson Bay in terms of ice thickness and sea surface temperature (Saucier, 2014), with counter-clockwise ocean currents leading to thicker ice cover along the eastern shore of the bay. Quaqtaq is located in Hudson strait, where wind and air temperature patterns are different from those in Hudson Bay, and pressured ice is common.

315 Freeze-up/break-up date predictions of the models at 30 and 60 lead days versus observed dates are presented in Figures 11 and 12. The red line in each plot represents a perfect one-to-one prediction and the pink region shows the acceptable 7 days difference that will still be considered as a correct prediction according to the CIS criteria. The width of the pink zone on each plot varies as the total time frame of break-up and freeze-up at each location is different (i.e. the subplots have different x and y axes). In addition, the year of 2010 is omitted from these plots as it was an anomalously warm year (Hochheim and Barber,
320 2014).

For freeze-up, 30 lead days predictions are more concentrated and closer to the pink zone while there is more dispersion and outliers observed for 60 lead days predictions (Fig 11). In addition, Augmented model predictions have fewer outliers than Basic model predictions. For the port of Churchill, predictions are close to the center and inside or close to the pink zone for both models and both lead times compared to other locations. The Basic model especially at 30 lead days, predicts freeze-up

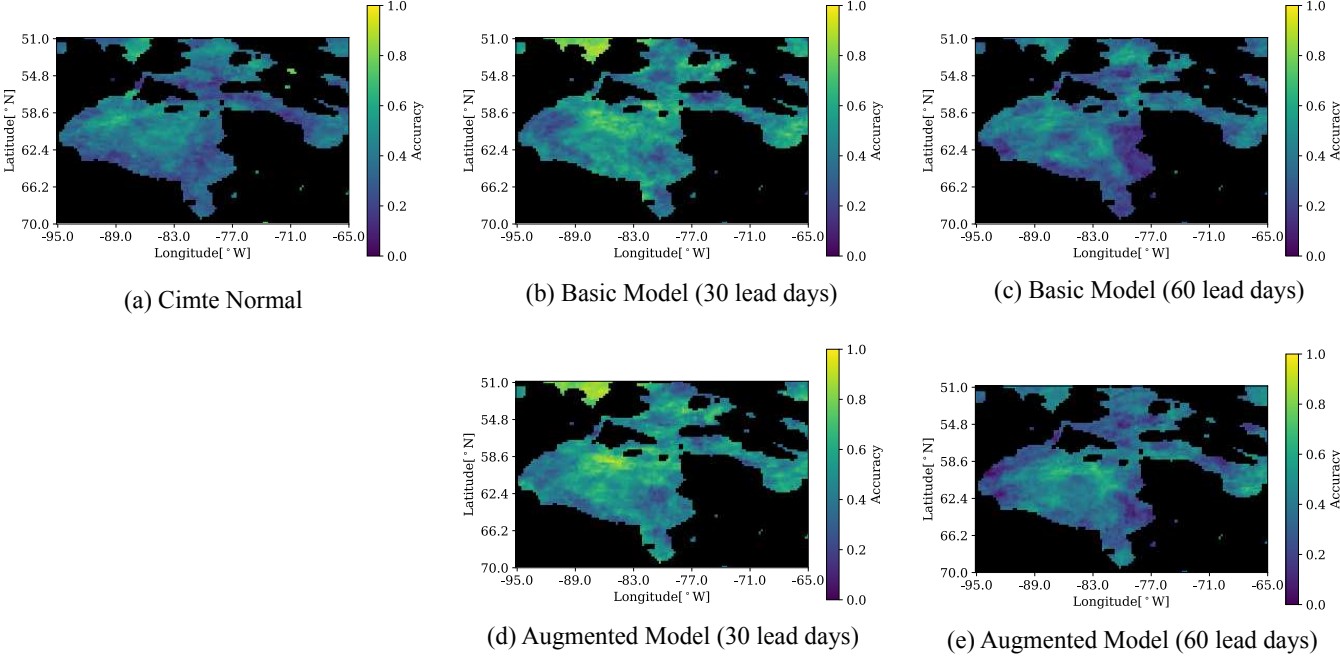

**Figure 9.** Accuracy of predicted break-up date within 7 days. Break-up dates are checked from May 1st to July 31st. The 7-day window is chosen to match to definition used by the Canadian Ice Service.

dates of several years with a consistent delay for Inukjuak while for Quaqtaq its predictions are earlier than observed dates. In Fig 12, similar to freeze-up, break-up dates are better captured by Augmented model at 30 lead days as compared to 60 lead days, where predictions are more scattered. Also, the patterns of early and delayed predictions are not as visible for break-up as for freeze-up for Inukjuak and Quaqtaq ports.

### 7.2.2 Freeze-up and Break-up in comparison with operational ice charts

To assess the operational capability of the models, it is important to consider the authoritative source of information used by shipping operators as a baseline for comparison, which is operational ice charts. Three sites were selected for this assessment: Kivalliq polynya near Arviat port (61.19N, 93.49W), Kinngait (64.05N, 76.48W) and Sanirajak (68.83N, 81.10W). These sites were selected because each is both near a port location and associated with a polynya; therefore the ice cover is challenging to predict. The accuracy of freeze-up and break-up dates at each site was evaluated against both CIS regional ice charts and ERA5 baselines. The predictions of the Basic and Augmented models at 30 and 60 lead days were assessed using the mean absolute error (MAE) and accuracy within 7 days. Median break-up and freeze-up dates derived from CIS regional ice charts from 1980 to 2010 and published in the Canadian Ice Service's Ice Atlas 1980-2010 (CIS, 2013) are also evaluated using the same methodology. For the Sanirajak site, each time break-up was outside the date range defined by the extraction methodology (May 1st to July 31st) from the ERA5 baseline or model forecast, the missing date was replaced by the ice atlas freeze-up date,

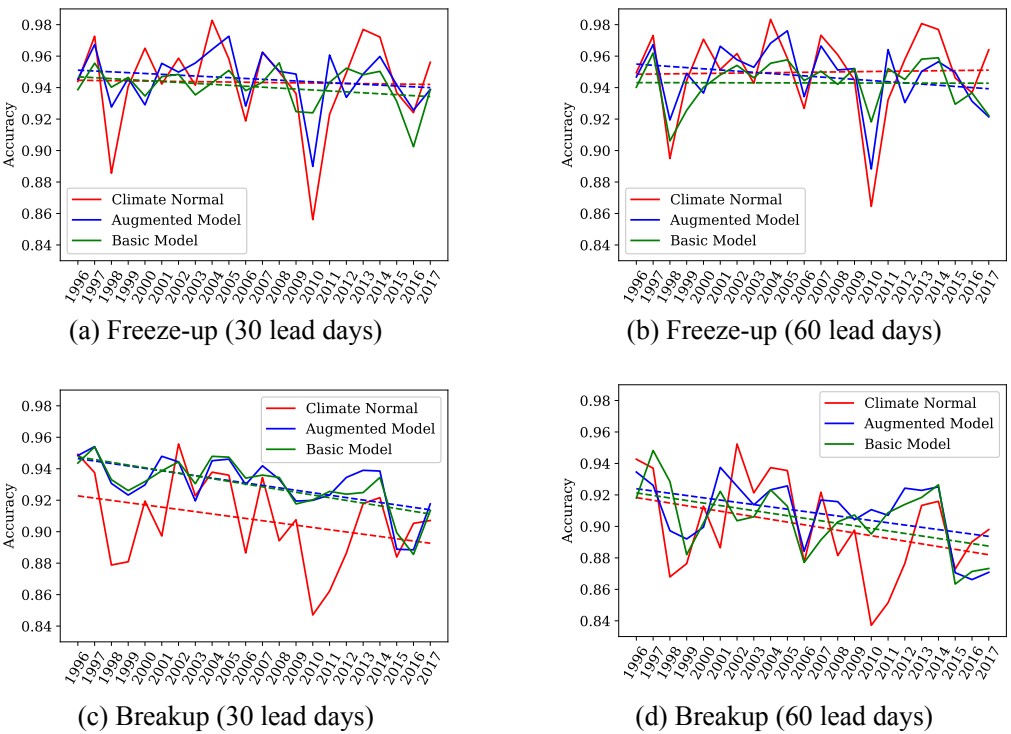

(a) Freeze-up (30 lead days)    (b) Freeze-up (60 lead days)

(c) Breakup (30 lead days)    (d) Breakup (60 lead days)

**Figure 10.** Accuracy of the Climate Normal, Basic model and Augmented model freeze-up and break-up predictions over the years at 30 and 60 lead days. Dashed lines show the trend.

|  |  | Break-up mean absolute error (days) | | | | | Break-up Accuracy | | | | |
|---|---|---|---|---|---|---|---|---|---|---|---|
|  |  | 30 lead days | | 60 lead days | | | 30 lead days | | 60 lead days | | |
|  | Baseline | Basic | Aug | Basic | Aug | Ice Atlas | Basic | Aug | Basic | Aug | Ice Atlas |
| Kivalliq | ERA5 | 14.10 | 12.38 | 13.67 | 18.14 | 15.52 | 0.29 | 0.48 | 0.38 | 0.10 | 0.43 |
| Kinngait | ERA5 | 13.05 | 13.14 | 17.00 | 13.81 | 17.71 | 0.14 | 0.29 | 0.38 | 0.24 | 0.29 |
| Sanirajak | ERA5 | 25.19 | 24.24 | 20.76 | 17.52 | 87.33 | 0.43 | 0.48 | 0.38 | 0.48 | 0.00 |
| Kivalliq | Ice Chart | 10.10 | 13.52 | 14.52 | 15.57 | 13.90 | 0.43 | 0.19 | 0.29 | 0.19 | 0.29 |
| Kinngait | Ice Chart | 25.76 | 25.86 | 28.95 | 25.57 | 34.62 | 0.05 | 0.05 | 0.10 | 0.14 | 0.05 |
| Sanirajak | Ice Chart | 90.90 | 97.95 | 86.67 | 82.48 | 18.67 | 0.05 | 0.05 | 0.05 | 0.00 | 0.43 |

**Table 1.** Break-up mean absolute error and accuracy at selected sites using data from the CIS Ice Atlas, Basic and Augmented (Aug.) models at 30 lead days and 60 lead days versus baseline observations derived from CIS regional ice charts and ERA5.

340    October 22nd, in order to calculate both metrics. This was done in order to handle multi-year ice situation when no break-up dates are available.

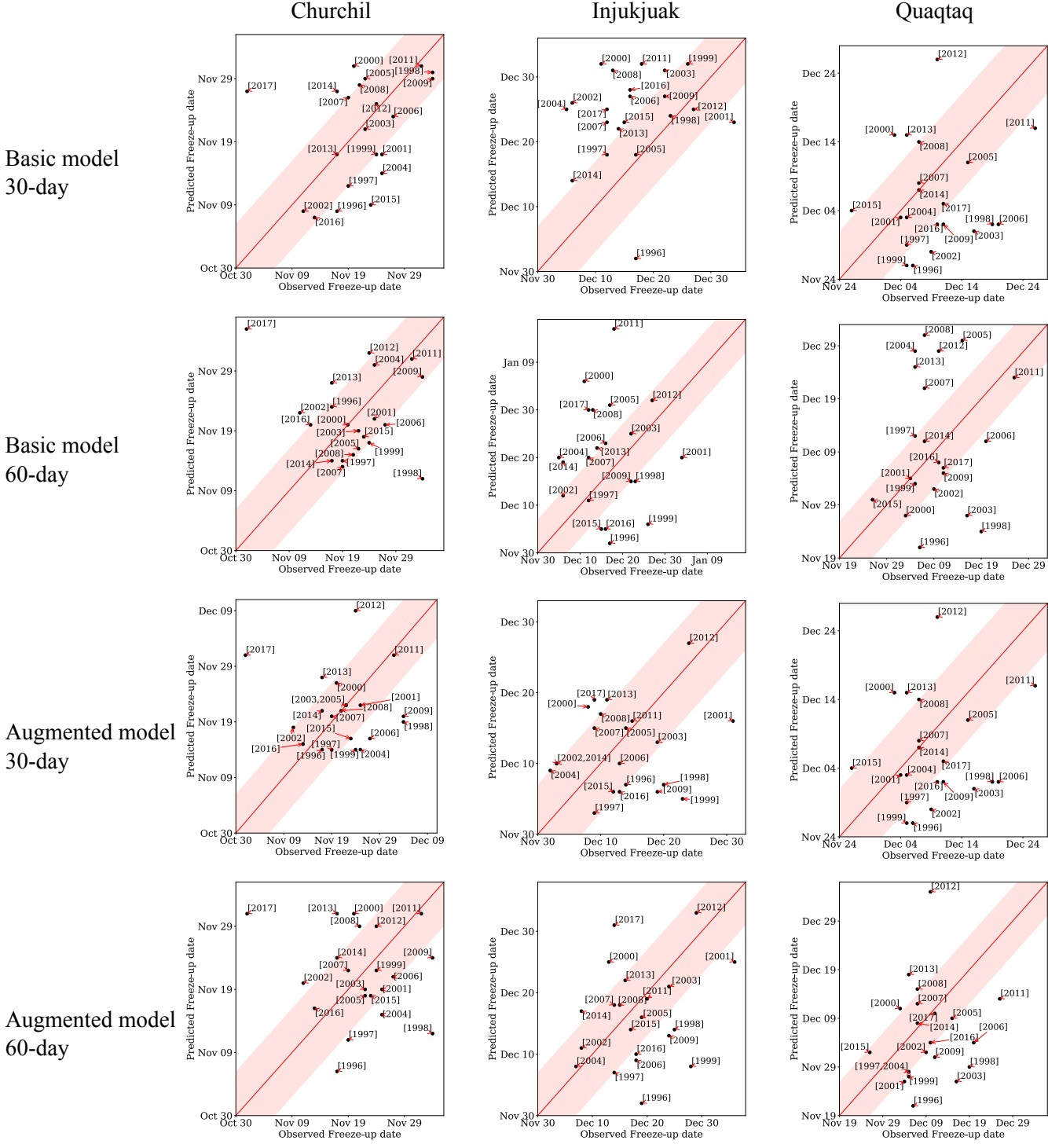

**Figure 11.** Comparison between forecast and observed freeze-up dates at pixels located in the vicinity of ports in the study region for 30 lead days and 60 lead days. Each dot represents one year. The red line represents perfect predictions and the pink area represents ± 7 days of the red line, which is commonly assumed as an acceptable error range.

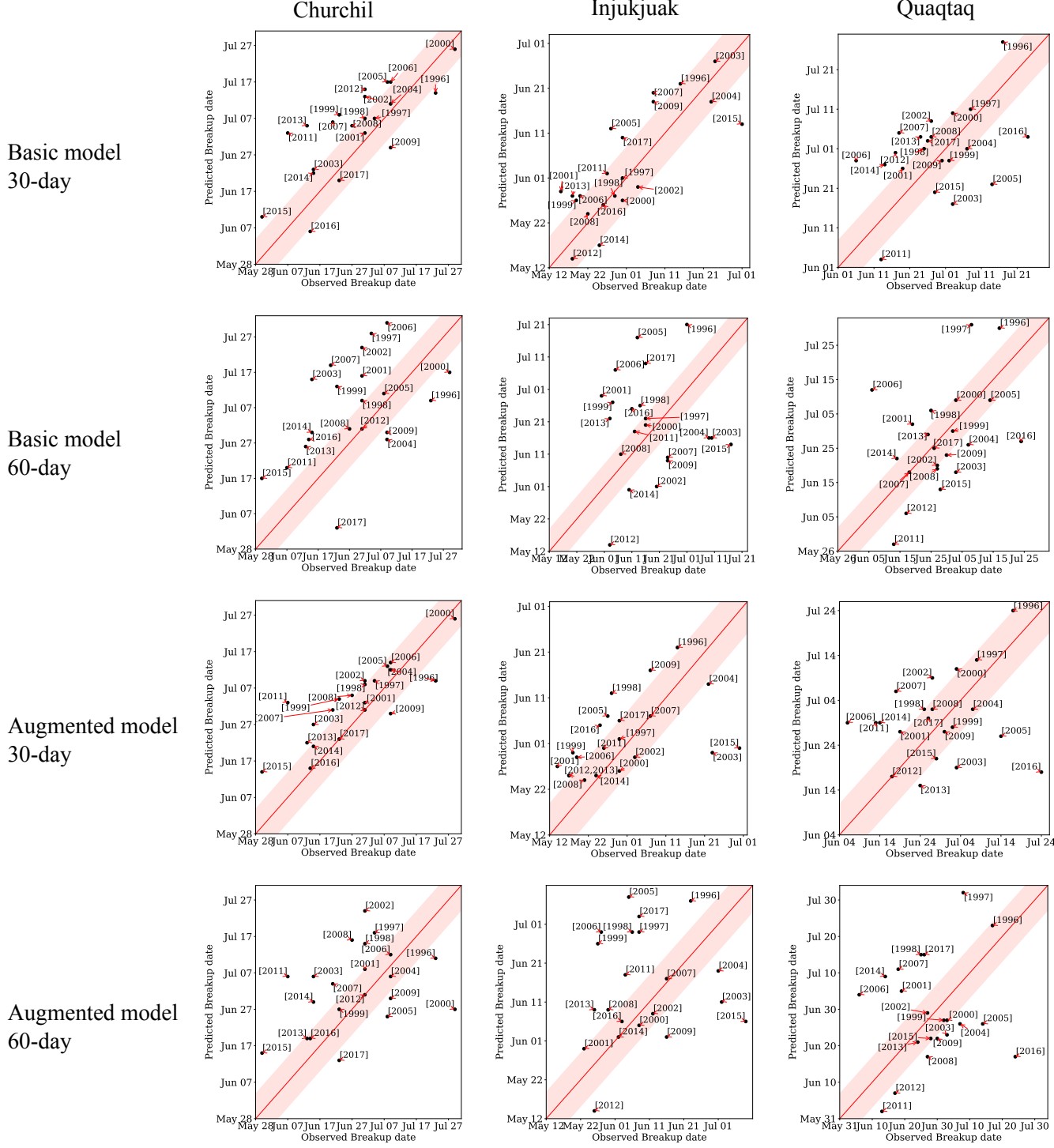

**Figure 12.** Comparison between forecast and observed break-up dates at pixels located in the vicinity of ports in the study region for 30 lead days and 60 lead days. Each dot represents one year. The red line represents perfect predictions and the pink area represents ± days of the red line, which is commonly assumed as an acceptable error range.

| | | Freeze-up mean absolute error (days) | | | | | Freeze-up Accuracy | | | | |
|---|---|---|---|---|---|---|---|---|---|---|---|
| | | 30 lead days | | 60 lead days | | | 30 lead days | | 60 lead days | | |
| | Baseline | Basic | Aug | Basic | Aug | Ice Atlas | Basic | Aug | Basic | Aug | Ice Atlas |
| Kivalliq | ERA5 | 6.05 | 7.71 | 5.81 | 7.95 | 5.33 | 0.62 | 0.52 | 0.71 | 0.52 | 0.71 |
| Kinngait | ERA5 | 7.48 | 12.38 | 11.00 | 13.33 | 8.10 | 0,62 | 0.43 | 0.38 | 0.33 | 0.71 |
| Sanirajak | ERA5 | 9.10 | 6.48 | 9.86 | 8.52 | 8.14 | 0.43 | 0.67 | 0,43 | 0.52 | 0.62 |
| Kivalliq | Ice Chart | 8.90 | 9.90 | 9.81 | 10.81 | 7.81 | 0.48 | 0.52 | 0.43 | 0.52 | 0.62 |
| Kinngait | Ice Chart | 13.90 | 19.67 | 18.10 | 20.43 | 13.76 | 0.43 | 0.33 | 0.29 | 0.33 | 0.52 |
| Sanirajak | Ice Chart | 10.33 | 15.52 | 15.67 | 17.00 | 9.86 | 0.43 | 0.33 | 0.29 | 0.24 | 0.43 |

**Table 2.** Freeze-up mean absolute error and accuracy at selected sites using data from the CIS Ice Atlas, Basic and Augmented (Aug.) models at 30 lead days and 60 lead days versus baseline observations derived from CIS regional ice charts and ERA5.

For break-up, at the Kivalliq site, the Augmented model at 30 lead days showed the best performance based on MAE and accuracy metrics using ERA5 baseline, while the Basic model at 30 lead days tends to perform better using CIS ice charts as baseline (Table 1). The break-up dates for the Kinngait site have higher interannual variability, reflected by the poor performance of the ice atlas at this site for both baselines. At the Kinngait site the break-up forecast skill is relatively consistent for both Basic and Augmented models at 30 and 60 lead days, but shows higher skill using the ERA5 baseline compared to the ice charts. The difference in break-up dates derived from the two baselines is significant at this site (Table 3), where events such as early break-up in March 2012 are captured by the ice charts but not by the ERA5 reanalysis. For Sanirajak site, the difference between baselines is exacerbated. The ice charts consistently detect early break-up at this site, while the ERA5 reanalysis does not. For example, the majority of break-up dates derived from ice charts were before July 1st, whereas this occurs only once in 2016 using the ERA5 baseline. The baseline discrepancy explains why Basic and Augmented models performed better using ERA5 baseline, while the ice atlas has better skill using the CIS ice chart baseline. However, both have similar skill using their corresponding baseline.

For freeze-up, at the Kivalliq site, all models perform well, with the lowest freeze-up accuracy at 0.52 using ERA5 baseline or 0.43 using CIS ice charts baseline (Table 2). For all sites, the ice atlas showed the highest freeze-up forecasting skill against both baselines, due to the lower interannual variability in freeze-up dates compared to break-up dates. These results are consistent with the freeze-up and break-up accuracy maps (Figs 8 and 9).

Table 3 highlights the discrepancy between the two baselines using the same metrics as Tables 1 and 2, where MAE can be interpreted as Mean Absolute Difference between the two baselines and the accuracy can be interpreted as the fraction of the time the baseline dates are within 7 days of each other. As expected, there is a minimal discrepancy for large and uniform areas such as the Kivalliq site, explained by the weekly publication frequency of the CIS regional ice charts. The discrepancy is higher for smaller and localized polynyas, such as the Kinngait and Sanirajak sites, where the low resolution passive microwave

instruments used by ERA5 do not detect them compared to CIS regional ice charts, which rely in part on higher resolution SAR data.

| | Break-up | | Freeze-up | |
|---|---|---|---|---|
| | MAE | Accuracy | MAE | Accuracy |
| Kivalliq | 8.76 | 0.48 | 5.81 | 0.76 |
| Kinngait | 18.52 | 0.38 | 8.14 | 0.67 |
| Sanirajak | 85.52 | 0.10 | 11.33 | 0.48 |

**Table 3.** Discrepancy between break-up and freeze-up dates derived from ERA5 and CIS regional ice charts. MAE refers to the mean absolute error (days). Accuracy is the fraction of freeze-up or break-up events for which the baseline dates are within 7 days of each other.

### 7.3 Comparison with forecast data from ECMWF S2S system

To evaluate our approach further, binary accuracies are calculated using sea ice concentration from the ECMWF S2S system as the baseline for comparison. Results are shown only for forecasts launched during months for which there are notable differences between the methods, which are May-June and October-November. Figure 13 shows that during May and June both the Basic and Augmented models have a higher binary accuracy than the S2S forecasts, while during October and November the opposite behaviour is observed, with the Basic and Augmented models having similarly low accuracies. We investigate these differences using false positive and false negative rates. The false positive rate is $\mathcal{FP}_{rate} = \mathcal{FP}/(\mathcal{FP} + \mathcal{TN})$ and is the ratio of the number of days for which water is incorrectly classified as ice (false positives, $\mathcal{FP}$) to total number of days classified as water ($\mathcal{FP} + \mathcal{TN}$), where $\mathcal{TN}$ is the true negatives, or number of days correctly classified as water. The false negative rate is $\mathcal{FN}_{rate} = \mathcal{FN}/(\mathcal{FN} + \mathcal{TP})$, and is the ratio of the number of days for which ice is incorrectly classified as water (false negative, $\mathcal{FN}$) to the total number of days classified as ice ($\mathcal{FN} + \mathcal{TP}$), where $\mathcal{TP}$ is the true positives, or number of days correctly classified as ice. Recall the observation used is the thresholded sea ice concentration from ERA5.

Figure 14 shows spatial maps of the binary accuracy, false positive rates and false negative rates, for forecasts at 30 lead days launched on dates between June 1st and June 30th. These forecasts correspond to ice conditions from July 1st to July 30th. The white regions correspond to locations masked out due to land, or where there are no positives (days classified as ice) in the false positive plots, and similarly in the false negative plots. For example, there is no ice in the northwest portion of the domain at this time of the year. For the Basic and Augmented models there is a high false positive rate in the south-east portion of Hudson Bay, indicating the sea ice is not retreating fast enough relative to the observations. However, for the S2S forecasts the false positive rate is high over almost all of Hudson Bay, including Hudson Strait. Climate Normal has the lowest false positive rate of the approaches examined here. For the false negative rate, different behaviour is observed with a high false negative rate for Climate Normal, indicating too much open water, and slightly lower false negative rates for the Basic and Augmented models. The Augmented model has a higher false negative rate than the Basic model, suggesting some of the overprediction of open water may be related to the additional air temperature or windspeed data that are input to this model. The strong recovery

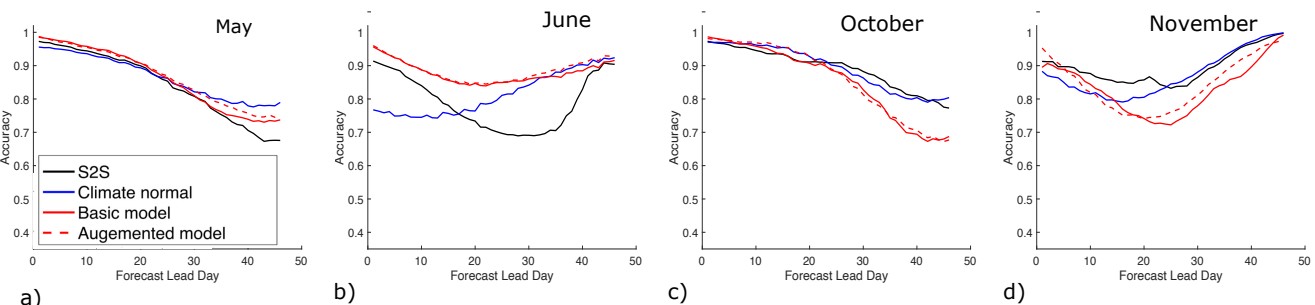

**Figure 13.** Binary accuracy as a function of lead day for forecasts launched in a) May; b) June; c) October; d) November. These months were chosen because they display the largest differences between the various forecasting methods. Binary accuracies are evaluated using data from both 2016 and 2017.

**Figure 14.** Binary accuracy, false positive rate and false negative rate calculated using data from forecasts launched between June 1-30, 2016 and June 1-30, 2017 at 30 lead days. These correspond to conditions from July 1-30, 2016 and July 1-30, 2017.

of the binary accuracy of the S2S forecasts around day 35 (Fig 13b) is due to the ice quickly retreating in these forecasts (not shown).

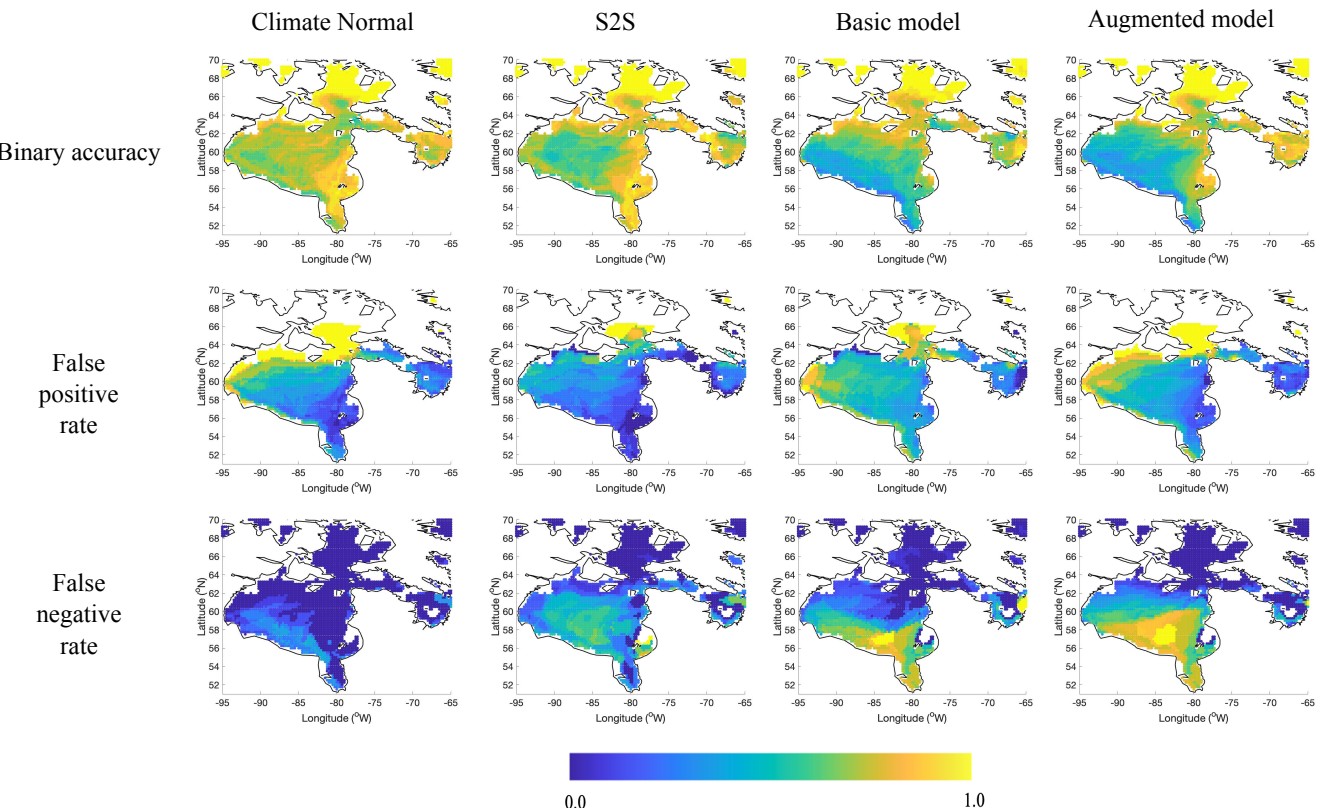

**Figure 15.** Binary accuracy, false positive rate and false negative rate calculated using data from forecasts launched between October 1-31, 2016 and October 1-31, 2017 at 45 lead days. These correspond to conditions from November 15-December 16, 2016 and November 15-December 16, 2017.

During October and November the S2S forecasts have a much higher binary accuracy than those from the Basic and Augmented models (Fig 13 c, d). The poor performance of the Basic and Augmented models is in part due to the opening of the Kivalliq polynya (Bruneau et al., 2021) in northwestern Hudson Bay (Fig 15, false positive rates). This is a large latent heat polynya that is sustained in part due to strong offshore winds. The Basic model, Augmented model and Climate Normal all predict freeze-up too quickly in this region, in comparison to the observations, while the S2S forecasts are able to represent

this better (false positive rates, Fig 15). The superior performance of the S2S forecasts may be because the S2S system uses a prognostic sea ice model coupled to the atmosphere. When ice starts to form, this reduces the heat exchange from the ocean to the atmosphere, and the rate of ice growth slows. The proposed approach may have had trouble representing this phenomena because the associated patterns may not have been represented in the training data.

## 8 Discussion

The proposed spatio-temporal sea ice forecasting method is capable of predicting sea ice presence probabilities with skill during May, June and July (break-up) in comparison to both Climate Normal and sea ice concentration forecasts from a leading S2S system (Fig 4 d,e and Fig 13 a,b). Results during freeze-up are more mixed, with an indication of higher accuracy in November in comparison to Climate Normal at short lead times (Fig 4 d,e), but degradation at longer lead times, and larger discrepancies with S2S forecasts (Fig 13 c,d). Regarding the poor performance of the Basic model in predicting freeze-up at 30 lead days

(Fig 8b) vs 60 lead days (Fig 8c), we note the freeze-up criteria is checked for dates between October 1st and January 31st. For this range of dates, 30-day forecasts would have been launched between September 1st and December 31st, trained on data from August 1st to October 31st (for the September model) and November 1st to January 31st (for the December model). In contrast, 60-day forecasts would have been launched one month earlier, and trained on data covering the same three month span. We hypothesize the 60-day forecasts are better than the 30-day forecasts because the air temperature can have more of

an impact for 60-day forecasts as the open water season is considered more heavily in the training data for the 60-day model (training data extends into July). Hochheim and Barber (2014) note a dependence of sea ice extent on air temperatures during freeze-up in this central region of Hudson Bay. The additional inputs to the Augmented model, which includes air temperature and the wind components, may account for the improved performance of the Augmented model in comparison to the Basic model for this scenario.

Throughout the paper the Basic and Augmented models have been compared. While the Augmented model was not developed to address a specific problem with the Basic model, it was developed to incorporate a climate normal, which can help the model generalize, meaning produce better forecasts over a wider range of conditions. It was found (Fig 4f and trend lines in Fig 10) the Augmented model generally has higher accuracy than the Basic model. The comparison with the S2S forecasts and Climate Normal, shown in Fig 13, 14 and 15, indicate these two approaches are in better agreement with each other than the

S2S forecasts, with the Augmented model in closer agreement with Climate Normal that the Basic model, as expected.

It is worthwhile to consider how our results compare with those of similar studies in the region. Gignac et al. (2019) and Dirkson et al. (2019) developed methods for probabilistic forecasting based on fitting probability distribution functions (PDFs) to historical passive microwave sea ice concentration data. Gignac et al. (2019) in particular focused on the same geographic region as the present study, choosing a beta PDF to fit the data and define a model from which they could query the probability

of ice, given a date. Following the same definition of break-up and freeze-up as used here, they found their approach was able to capture freeze-up and break-up within one or two weeks of dates provided by the Canadian Ice Service (CIS) ice atlas, with the exception of Sanirajak (formerly known as Hall Beach), similar to the results reported here. Their discrepancy was 9 weeks (or 63 days), hence slightly shorter than ours (Table 1), although we have used ice charts directly, while they used a climatology based on ice charts. They related this discrepancy to the use of a mean when processing the passive microwave

data, in comparison to the median used for the ice atlas. We agree this could be a contributing factor, but also note the passive microwave data are biased when the ice is thin (Ivanova et al., 2015), as is the case in a polynya. Dirkson et al (2019) developed a related approach but used zero and one inflated beta distribution. Their PDF is fit to data from a prognostic modelling system,

CanSIPS, (Canadian Seasonal to Interannual Prediction System), which consists of two coupled atmosphere-ice-ocean models. A bias correction approach is applied to their predictions and CanSIPS output before comparison with observational data, which was provided by HadISST2 sea ice and surface temperature dataset. Their predictions show skill in Hudson Bay for forecasts initialized in May and June for 1-2 months (their Fig 10), but little skill for freeze-up, similar to what is found in the present study. Studies using coupled ice-ocean models (Sigmond et al., 2016; Bushuk et al., 2017) show more skill for freeze-up than for break-up, consistent with the S2S results found here.

## 9 Conclusion

This study has focused on sea ice presence probability forecasting using deep learning methods at a daily time scale with lead times up to 90 days. The Basic model uses eight input variables from the ERA5 dataset for the 3 days prior to the forecast launch date. An augmented version of this Basic model is also proposed which takes an additional input from Climate Normal. Comparing the binary accuracy of the Basic and Augmented models and Climate Normal demonstrated improvements of up to 10% relative to Climate Normal for both the break-up and freeze-up seasons, especially for short lead times (up to 30 days). The probability assessment by the calibration analysis (Fig 6) and Brier score also revealed most differences in break-up and freeze-up season with scores from Augmented model slightly better in comparison to the Basic model. The analysis of break-up and freeze-up date prediction of the models shows that the Augmented model is more capable of accurately predicting these dates within 7 days compared to the Basic model while the accuracy of both models degrades with increasing lead time. It should be noted that both models show substantial improvement over Climate Normal at 30 lead days for break-up date prediction.

The model is demonstrated in hindcast mode here, but it is intended to be used for forecasting. Compared to dynamical forecasting systems in this domain, the proposed approach has the advantage of time efficiency as once the initial model is trained, the fine-tuning process for new inputs (consisting of one year of training data) takes around 15 minutes on a Tesla GPU and each inference takes around 10 seconds to complete. We also do not envision it to be difficult to use our approach with alternate input data from the point of view of model architecture. We recommend that if one was to use input data from a different source that they fine-tune the existing weights to account for the different data dependencies in the input data (in particular consider that only a subset of model variables are used, dependencies present in one subset may be partially considered in a different subset for a different model).

A limitation of our approach is that it relies on data from reanalyses. Without an additional downscaling module, the spatial resolution of our forecasts cannot exceed that of the input data, which here is 31 km. We note this resolution is similar to that used in other studies on seasonal forecasting that have been developed with mariners in mind. For example, passive microwave data were used for development of the probabilistic approach of Gignac et al. (2019) and for validation of subseasonal-to-seasonal sea ice predictions (Zampieri et al. 2018). While passive microwave sea ice concentration data are often gridded to 25 km, the spatial resolution of the brightness temperature data used to generate the sea ice concentration are typically coarser. The 19.35 GHz channel on the SSMI and SSMI/S sensors (often used to produce sea ice concentration observations) has an

instrument field of view of approximately 45 km x 69 km. The spatial resolution used here is similar to that used in studies that carry out seasonal forecasting using a dynamic ice-ocean model (or similar) where a sea ice state vector is predicted as a function of time (Sigmond et al., 2016, Askenov et al. 2017). Hence, in terms of spatial resolution, the ML approach proposed in this study is not coarser than other commonly used approaches, some of which target marine transportation.

As future work, we plan to expand the experiments over the entire Arctic region, and deploy ensemble methods using more recent deep learning architectures. Looking into possible improvements by adding SIC anomaly as additional input variable as investigated by Kim et al. (2020) is another path to explore.

*Code and data availability.*    Code availability

The ECMWF ERA-5 atmospheric reanalysis data (Hersbach et al., 2018) are available at
https://cds.climate.copernicus.eu/cdsapp/dataset/reanalysis-era5-single-levels?tab=overview. The subseasonal to seasonal forecasting data used for comparison (Vitart et al., 2018) are available at
https://apps.ecmwf.int/datasets/data/s2s-realtime-daily-averaged-ecmf/levtype=sfc/type=cf/.

The ice atlas data are available at, Canadian Ice Service, Ice Atlas 1980-2010: https://publications.gc.ca/site/eng/441147/
publication.html, while ice charts are available from the Canadian Ice Service archive, Regional Ice Charts, https://iceweb1.
cis.ec.gc.ca/Archive/.

The model source code can be downloaded from the repository website https://github.com/zach-gousseau/sifnet_public or https://zenodo.org/record/6855080 See the project website's README.md for details.

*Author contributions.*    PL, MK and MR designed and initiated the study and proposed the model. NA designed the experimental setup and performed the simulations and analysis of the results. PL and KAS supervised the study and provided feedback. NA, PL, MK, and KAS contributed to the development and writing of this paper.

*Competing interests.*    The authors declare that they have no conflict of interest.

*Acknowledgements.*    The authors would like to acknowledge funding from the National Research Council of Canada through the AI4Logistics
and Ocean Programs and computing resources provided by Compute Canada. The ERA5 data were downloaded from CDS website. The results contain modified Copernicus Climate Change Service information 2022. Neither the European Commission nor ECMWF is responsible

for any use that may be made of the Copernicus information or data it contains. We would like to thank Zacharie Gousseau for putting together the code repository. We are also grateful for insightful comments offered by the reviewers.

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
