# Peer review of "Probabilistic Spatio-temporal Seasonal Sea Ice Presence Forecasting using Sequence-to-Sequence Learning and ERA5 data in the Hudson Bay region"

_The Cryosphere, 2021_

## Referee Comment (RC1)

**Evaluation: Probabilistic Gridded Seasonal Sea Ice Presence Forecasting using Sequence to Sequence Learning**

**Overall evaluation**

| Principal criteria | Excellent (1) | Good (2) | Fair (3) | Poor (4) |
|---|---|---|---|---|
| *Originality (novelty)* | | X | | |
| *Scientific quality (rigour)* | | | X | |
| *Significance (impact)* | | X | | |
| *Presentation quality* | | | X | |

**TC review questions**

1. **Does the paper address relevant scientific questions within the scope of TC?**
   Yes, absolutely.

2. **Does the paper present novel concepts, ideas, tools, or data?**
   The manuscript uses innovative approaches in ice forecasting.

3. **Are substantial conclusions reached?**
   More work needs to be done on this matter. Even though the tool is solid and technically pertinent, more has to be said on the overall conclusions of its applications and usefulness.

4. **Are the scientific methods and assumptions valid and clearly outlined?**
   On this point, more clarity must be achieved in the manuscript. Research hypotheses and objectives could be more precisely outlined. Details about the different tests and configurations should be brought to the attention of the readers.

5. **Are the results sufficient to support the interpretations and conclusions?**
   Details are lacking in this matter. A lot of attention has been put on the technical matters but too few on the applications of the model itself. This comparison and "real world application test" part will prove the pertinence of the tool, not only the technical justifications.

6. **Is the description of experiments and calculations sufficiently complete and precise to allow their reproduction by fellow scientists (traceability of results)?**
   No. More details need to be provided. Especially the ML architecture and configuration that needs to be explained and also, supported by a schematic representation.

7. **Do the authors give proper credit to related work and clearly indicate their own new/original contribution?**

   More could be done, especially in terms of comparison with existing models. However, when speaking about the model and ML approaches, the authors gave proper credit to existing work and clearly stated their contribution to the field.

8. **Does the title clearly reflect the contents of the paper?**

   Not entirely, it could be more precise. At first glance, I thought it was based on remote sensing time series and had no idea of what "seasonal" referred to.

9. **Does the abstract provide a concise and complete summary?**

   It is quite short and could provide more details. More could be said about the applicability and practicality of the developed tool as well as about the limitations.

10. **Is the overall presentation well structured and clear?**

    Yes. The authors tend to go straight to the point when needed.

11. **Is the language fluent and precise?**

    Yes absolutely.

12. **Are mathematical formulae, symbols, abbreviations, and units correctly defined and used?**

    Yes.

13. **Should any parts of the paper (text, formulae, figures, tables) be clarified, reduced, combined, or eliminated?**

    Both methodology and discussion should go into more details. Especially the discussion (see comments).

14. **Are the number and quality of references appropriate?**

    There are little to no reference listed about the known Hudson Bay sea ice spatiotemporal patterns and dynamics. As I suggested more application examples and comparison with existing methods and data, I suppose a few references will be appended.

15. **Is the amount and quality of supplementary material appropriate?**

    Since it is not the final submitted version, I understand that at this point the code isn't shared. However, this would be an excellent asset to help readers and potential users appreciate the authors work.

**Reviewer comments**

This manuscript presents an innovative forecast tool for sea ice conditions (presence) based on a machine learning approach using sequence to sequence learning for both short term (7 days) and long term (up to 90 days) predictions.

The presented tool is, without a doubt, something that is of interest to the sea ice expert community and is based on novel methods of machine learning that make analysis of massive information datasets possible nowadays.

Even though the pertinence of the presented tool, several improvements must be done on the manuscript. The research design in itself has to be described into more details, especially by providing a more complete description of the different tests and protocols followed in the experiments and model calibration part.

In addition, many paragraphs, especially in the methodology part, could be supported by figures and schematic representations, for example, the ML design and architecture.

Also, as the Hudson Bay region is highly documented and studied, the results obtained by your approach could be compared to data provided in the sea ice atlas from the Canadian Ice Service or results from other probabilistic/modelling approaches applied on the Hudson Bay (Saucier et al. 2004, Hochheim and Barber 2014, Kowal et al. 2017, Gignac et al. 2018, Dirksen et al. 2021.). Even though the comparison may not be quantitative, a qualitative assessment, outlining the differences between the approaches and the strategic advantages you provide using ML would be relevant.

Overall, the presented method and tools appear as scientifically sound and clear. They should, however, be described more carefully and more examples of applications of the model shall be presented to the readers.

It is a work of great interest and I hope my comments will guide and help you in improving your manuscript.

**Major comments**

1. As aforementioned, _comparisons have to be made and applications examples provided_. Especially in areas of high variability or in the presence of particular entities such as polynyas or narrower Bays (Frobisher Bay or Hall Beach polynya, for example).
2. _Limitations of the approach shall be discussed_. The model is providing forecasts on a ~31 km grid. How does this affect the usage capabilities for the principal expected users (the mariners)? This should definitely be discussed more in depth.
3. _Sensibility to the input "sea ice normal condition" wasn't discussed_, when speaking about the augmented model. _Were variable time spans tested?_ If so, did they generate similar forecasts? If not, how would you explain this situation? In other words, a certain "sensibility analysis" would be convincing about the model capabilities.
4. I strongly suggest that you _add a map of your validation sites_ and provide a short description of each. For example, Quaqtaq is located in a bay, narrower than 31km. How this does affects the results? Also, _why were these 3 sites chosen_? Have you found any irregularities in the ERA-5 sea ice concentration dataset you used to calibrate your model?
5. Nowhere in the manuscript have I found a _justification on why it is the presence of ice that is modeled and not the concentration_. That should be discussed since the OSI-SAF OSI409 (which are SIC) data are ingested into ERA-5.

**Specific comments**

- Line 10 : Define "high spatial resolution" for the reader. Depending on your field, it differs.
- Line 46 : Define sea ice presence (SIC > 15%).

- Line 51 : This information should be provided way before, otherwise, some will think, as I did, that you use remote sensing data.
- Line 57 : Why not starting in 1979 ?
- Line 67 : Remove last "and".
- Line 87 : A schematic representation of the encoder and decoder parts would be useful.
- Line 92 : What time bin(-s) were used as input (12:00, 00:00, a daily average ?)
- Line 112 : How does extending to a longer input affects the forecast quality ?
- Line 126 : Can you describe these "extensive experimentations" ?
- Line 129 : "Chosen to be 10 years". Why is it so ?
- Line 135 : This processing logic should definitely be represented in a figure.
- Line 166 – 167 : The end of the sentence doesn't make sense. Consider reformulating.
- Line 181 : It seems counterintuitive. Can you explain why ?
- Figure 1 : Y-Axes for subfigures d-e-f should be Accuracy differences or ΔAccuracy.
- Line 220 : What would you link the lower accuracy in "central region" ? Is it the higher variability of the freeze-up pattern or to climate variables that are, given the distance to stations, less reliable in such areas ?
- Lines 236, 246 & 249 + Figures 8 & 9 : From what I know, it should be written Quaqtaq, not Quataq (https://www.makivik.org/quaqtaq/).

**References**

Dirkson, Arlan, et al. "Development and Calibration of Seasonal Probabilistic Forecasts of Ice-free Dates and Freeze-up Dates." Weather and Forecasting 36.1 (2021): 301-324.

Gignac, C., Bernier, M., & Chokmani, K. (2019). IcePAC–a probabilistic tool to study sea ice spatio-temporal dynamics: application to the Hudson Bay area. The Cryosphere, 13(2), 451-468.

Hochheim, K. P. and Barber, D. G.: An update on the ice climatology of the Hudson Bay system, Arct. Antarct. Alp. Res., 46, 66–83, 2014.

Kowal, S., Gough, W. A., and Butler, K.: Temporal evolution of Hudson Bay Sea Ice (1971–2011), Theor. Appl. Climatol., 127, 753–760, 2017.

Saucier, F., Senneville, S., Prinsenberg, S., Roy, F., Smith, G., Gachon, P., Caya, D., and Laprise, R.: Modelling the sea ice-ocean seasonal cycle in Hudson Bay, Foxe Basin and Hudson Strait, Canada, Climate Dynam., 23, 303–326, 2004.

---

## Author Comment (AC1)

**Reply to Reviewer 1- tc-2021-282 Asadi et al. 2021 - Probabilistic Gridded Seasonal Sea Ice Presence Forecasting using Sequence to Sequence Learning**

We sincerely thank the reviewer for the thorough review and excellent comments.
Reviewer comments are shown in black, our responses are shown in blue.

This manuscript presents an innovative forecast tool for sea ice conditions (presence) based on a machine learning approach using sequence to sequence learning for both short term (7 days) and long term (up to 90 days) predictions.
The presented tool is, without a doubt, something that is of interest to the sea ice expert community and is based on novel methods of machine learning that make analysis of massive information datasets possible nowadays.

Even though the pertinence of the presented tool, several improvements must be done on the manuscript. The research design in itself has to be described into more details, especially by providing a more complete description of the different tests and protocols followed in the experiments and model calibration part.
In addition, many paragraphs, especially in the methodology part, could be supported by figures and schematic representations, for example, the ML design and architecture.

We have included preliminary figures (shown later in this response) describing the training and testing sequence (model calibration) and also the ML architecture (encoder and decoder).

Also, as the Hudson Bay region is highly documented and studied, the results obtained by your approach could be compared to data provided in the sea ice atlas from the Canadian Ice Service or results from other probabilistic/modelling approaches applied on the Hudson Bay (Saucier et al. 2004, Hochheim and Barber 2014, Kowal et al. 2017, Gignac et al. 2018, Dirkson et al. 2021.). Even though the comparison may not be quantitative, a qualitative assessment, outlining the differences between the approaches and the strategic advantages you provide using ML would be relevant.

Thank you for this comment. Gignac et al (2019) and Dirkson et al (2019) developed methods for probabilistic forecasting of sea ice concentration based on fitting probability distribution functions (PDFs) to historical passive microwave sea ice concentration data. Gignac et al (2019) used a beta PDF. They found their approach was able to capture freeze-up and break-up within one or two weeks of dates provided by the Canadian Ice Service (CIS) ice atlas, with the exception of Sanirajak (Hall Beach), where there is a polynya. Dirkson et al (2019) developed a related approach but used zero and one inflated beta distribution. This distribution allows the endpoints of their probability distribution to be captured differently than the interior points. Their PDF is fit to output from a prognostic modelling system, CanSIPS, (Canadian Seasonal to Interannual Prediction System), which consists of two coupled atmosphere-ice-ocean models. A novel bias correction approach is applied to their predictions and CanSIPS output before comparison with HadISST2 sea ice and surface temperature dataset. Their predictions show skill in Hudson Bay for forecasts initialized in May and June for 1-2 months (their Fig 10). Building on this, a related approach was used in Dirkson et al (2021), where a PDF was fit to freeze-up and break-up dates. In that case, before bias

correction the model (CanSIPS) predictions of freeze up and break up in Hudson Bay were biased by 4 weeks, and 1-2 weeks respectively, where the data used for comparison is passive microwave sea ice concentration. The proposed bias correction methods (Dirsken et al. 2021) lead to improved prediction of these events in Hudson Bay for lead times of 1-6 months, depending on the location. For both of these studies the horizontal grid resolution was 100 km and time scales were monthly.

We also found two other relevant studies, those by Bushuk et al (2017) and Sigmond et al. (2016). Bushuk et al. (2017) use a fully coupled atmosphere-ice-ocean-land model with horizontal grid resolution for the ice-ocean component of 1 degree. They compute the anomaly correlation coefficient (ACC) of detrended sea ice extent from their model with that from passive microwave data and compare with a model of anomaly persistence. Their model shows skill relative to persistence in Hudson Bay, in particular in summer months, with lead times of 3-8 months. Sigmond et al. (2016) use CanSIPS. They examine anomaly correlation coefficients between sea ice advance and retreat dates computed from the model and from passive microwave sea ice concentration data. The grid resolution is relatively coarse (around 100 km), possibly because it is an ensemble forecasting system, which is computationally demanding. They find their model has skill (defined by statistically significant ACC) at lead times of 2-6 months (average 5) for sea ice advance (freeze-up) and 2-3 months (average 3) for sea ice retreat (break-up) (not detrended, results for Hudson Bay).

We will discuss these in more detail, bringing in context from Saucier et al (physical modelling) and Hocchiem and Barber and Kowal (trend analysis), in the revised manuscript.

It should be noted while our approach is similar those by Gignac et al. (2019) and Dirkson et al. (2019, 2021), in that it does not use dynamical sea ice models to produce a prediction, it is different in that it uses a sequence-to-sequence learning approach that is capable of producing forecasts. The present configuration is to used for 90-day forecasts, although it is evaluated in the submitted manuscript in hindcast mode to demonstrate the concept.

Overall, the presented method and tools appear as scientifically sound and clear. They should, however, be described more carefully and more examples of applications of the model shall be presented to the readers.

It is a work of great interest and I hope my comments will guide and help you in improving your manuscript.

Thank you very much. Your comments are very helpful.

**Major comments**
1. As aforementioned, *comparisons have to be made and applications examples provided*. Especially in areas of high variability or in the presence of particular entities such as polynyas or narrower Bays (Frobisher Bay or Hall Beach polynya, for example).

We are planning to bring in a comparison of freeze-up and break-up dates with ice charts for the ports identified in the manuscript (Churchill, Inukjuak and Quataq) as well as; Sanirajak (formerly

known as Hall Beach), and Frobisher Bay, where there are polynyas; and a location in the central part of Hudson Bay (away from shore). We will compare these freeze up and breakup dates to those in Gignac et al, 2019 (their Table 1). We have investigated using daily ice charts of the Canadian Ice Service for this purpose. These are sea ice concentration and ice type analyses derived through manual inspection of SAR data and other data available, with valid time of 18 UTC each day. However, these charts are not available on a daily basis through the entire ice season in the Hudson Bay region. Hence, our analysis will be based on regional ice charts, which are similar to daily ice charts, but bring in data over a week and have less fine spatial detail. We also considered using the CIS ice atlas for this comparison (as was used in Gignac et al) but the CIS ice atlas covers an earlier period. Note that Frobisher bay is not in our study region.

2. *Limitations of the approach shall be discussed*. The model is providing forecasts on a ~31 km grid. How does this affect the usage capabilities for the principal expected users (the mariners)? This should definitely be discussed more in depth.

The resolution is similar to that used in other studies on seasonal forecasting that have been developed with mariners in mind. For example, passive microwave data were used for development of the probabilistic approach of Gignac et al. (2019) and for validation of subseasonal-to-seasonal (S2S) time scales (Zampieri et al. 2018).  While passive microwave sea ice concentration data are often gridded to 25 km, the spatial resolutions of the brightness temperature data used to generate the sea ice concentration are typically coarser. The 19.35 GHz channel on the SSMI and SSMI/S sensors (often used to produce sea ice concentration observations) has an instrument field of view of approximately 45 km x 69 km (https://www.remss.com/missions/ssmi). For studies that carry out seasonal forecasting using a dynamic ice-ocean model (or similar) where a sea ice state vector is predicted as a function of time, our resolution is similar to that used in other approaches to be used to support (in part) marine transportation (Sigmond et al., 2016, Askenov et al 2017). Sigmond et al. (2016) assess seasonal forecasts of  sea ice advance and retreat at a spatial resolution of 100 km, while Askenov et al. (2017) examine navigation routes using a ¼ degree grid (nominally 28km, 9-14 km in Arctic) coupled sea-ice ocean model. For the latter study, their focus is on the Northern Sea route, and their study region does not include Hudson Bay or Hudson Strait. Nevertheless, the study by Askenov et al. (2017) study highlights potential future changes in Arctic ice cover that will lead increased navigability, but with significant risk

For trend analysis in the Hudson Bay region, which also supports planning activities, Hoccheim and Barber (2014) use passive microwave data (gridded to 25 km) to look at sea ice trends in the Hudson Bay system, while Andrews et al. 2018, use a combination of passive microwave data for offshore trends in ice cover, and regional ice charts from the Canadian Ice Service for nearshore trends.

3. *Sensibility to the input "sea ice normal condition" wasn't discussed*, when speaking about the augmented model. *Were variable time spans tested?* If so, did they generate similar forecasts? If not, how would you explain this situation? In other words, a certain "sensibility analysis" would be convincing about the model capabilities.

In the version submitted, the augmented model secondary input is based on climate normal computed as the average 2m air temperature (t2m), and average 10m wind components (u10 and v10), where the average is calculated from the first year (1985) until the last year of training (2015). We compared this method of computing the climate normal (CN1) with one that is based on the 10 years previous to the validation year (CN2). There was very little difference in the model predictions when these two different augmented models are used based on heat maps (of the type shown in Fig 1 in the submitted manuscript). This indicates the patterns our model is learning do not vary significantly over the 1985-2015 period in comparison to a decadal time scale. Examining the results on a year to year basis showed that when the shorter time period is used for climate model the year-to-year variability has a more significant impact on model performance, indicating climate normal is given less weight, or is more similar to the training data.

4. I strongly suggest that you *add a map of your validation sites* and provide a short description of each. For example, Quaqtaq is located in a bay, narrower than 31km. How this does affect the results?

We have put together a map of the port locations, including one requested at Sanirajak (formerly known as Hall Beach). The map of the study region is shown below with the port locations shown in red. The Insets show the port location (red) and the nearest point on the model grid (blue) that is outside of the land boundary (where landmask from ERA5 is less than 0.6), in addition to a bounding box that approximates a grid cell. The model grid point near Quaqtaq is located (correctly) in the water region because the landmask from ERA5 has a low value in that region due to the low elevation.

[Figure]

Also, *why were these 3 sites chosen*?
The sites were chosen because they represent locations with significantly different sea ice conditions. Churchill and Inukjuak are located on the east and west coasts of Hudson Bay, with Churchill being a major port as part of the potential Arctic Bridge shipping route. The east coast is significantly impacted by influx of freshwater inflow from rivers draining into Hudson Bay, while the

west coast region is impacted by northwesterly winds (there is a latent heat polynya, the Kivalliq polynya, that runs along the northwest shore of Hudson Bay). There are additionally east-west asymmetries in Hudson Bay in terms of ice thickness and sea surface temperature (Saucier, 2014), with counter-clockwise ocean currents leading to thicker ice covers along the eastern shore of the Bay. Quataq is located in Hudson strait, where wind and air temperature patterns are different from those in Hudson Bay, and pressured ice is common in the consolidated ice season. We will be bringing in Sanirajak and a location in the central part of the Bay (away from the coast) in the revised version.

Have you found any irregularities in the ERA-5 sea ice concentration dataset you used to calibrate your model?

We found there were irregularities with the ERA landmask file and the sea ice concentration. There were some locations indicated as land in the landmask file that had a non-zero sea ice concentration value. At these locations the sea ice concentration was set to zero. There were also some locations indicated as non-land in the landmask file that had a zero ice concentration, even when the ice concentration should be non-zero based on the atmospheric conditions and seasons. At these locations (indicated by landmask less than or equal 0.6), the sea ice concentration was set to the non-land average of neighboring pixels.

5. Nowhere in the manuscript have I found a *justification on why it is the presence of ice that is modeled and not the concentration*. That should be discussed since the OSI-SAF OSI409 (which are SIC) data are ingested into ERA-5

The approach used here predicts a grid of (uncalibrated) sea ice probabilities. For seasonal forecasting, probabilistic information is often desired (Gignac et al. 2019), in particular regarding freeze-up and break up (Wagner et. al., 2020). While the model does ingest sea ice concentration, this is combined with other environmental variables to produce a binary probability of ice vs water at the grid cell, or ice presence. This output is similar to probabilistic approaches by Gignac et al. (2019) and Dirkson et al (2019) where their model estimates the probability of sea ice concentration exceeding a certain threshold (e.g., 15%).

**Specific comments**
Line 10 : Define "high spatial resolution" for the reader. Depending on your field, it differs.

We were referring to the spatial resolution of 5-10 km. For example, the high-resolution ocean and sea ice forecasting system for the Arctic and North Atlantic oceans (Dupont et al., 2015).

Line 46 : Define sea ice presence (SIC > 15%).

We noticed that the term "sea ice presence" was first used at the beginning of the introduction, and have added the information there (sea ice concentration greater than 15%).

Line 51 : This information should be provided way before, otherwise, some will think, as I did, that you use remote sensing data.

The use of ERA5 has been added to the last paragraph of the introduction. The abstract has also been changed to explicitly mention this "Given the recent observations of the declining trend of Arctic sea ice extent over the past decades, seasonal forecasts are often desired. In this study machine

learning (ML) approaches are deployed to provide accurate seasonal forecasts based on ERA5 data as input"

Line 57 : Why not starting in 1979 ?

Initially the study was started with a different data set (other than ER5). These data started in 1985, hence that was used in this study as well because it provides a sufficient time series for training and testing.

Line 67 : Remove last "and". Corrected

Line 87 : A schematic representation of the encoder and decoder parts would be useful.

Below is a figure of the architecture which shows the encoder and decoder parts in more detail. The upper panel shows the overall architecture(described on lines 98-105 of the submitted manuscript, modified here).

*The overall architecture is shown in the figure below (panel a). The encoder starts by passing each daily sample through a feature pyramid network (Lin et al., 2017) so as to detect environmental patterns at both the local and large scales. Next, the sequence of feature grids extracted from the feature pyramid network are further processed through a convolutional LSTM layer (ConvLSTM) (Hochreiter and Schmidhuber, 1997; Xingjian et al., 2015), returning the last output state. This layer learns a single grid representation of the time series that also preserves spatial locality. Finally, the most recent day of historic input data is concatenated with the ConvLSTM output. The encoder provides as output a single raster with the same height and width as the stack of raster data input to the network, but with a higher number of channels such as to represent the fully encoded system state. The final encoded state is then fed to a custom recurrent neural network (RNN) decoder that extrapolates the state across the specified number of time-steps. It takes as input the encoded state with multiple channels and as output produces a state with the same height and width as the input over the desired number of time-steps in the forecast (here 90 days). Finally, a time-distributed network-in-network (Lin et al., 2013) structure is employed to apply a 1D convolution on each time-step prediction to keep the spatial grid size the same but reduce the number of channels to one, representing the daily probabilities of ice presence over the forecast period (e.g. up to 90 days).*

The lower panel shows the decoder (described on lines 106-113 of the submitted manuscript, modified here)

*The custom RNN decoder, as is common of many RNN layers, maintains both a cell state and a hidden state (Yu et al.,2019). First, the initial cell state and hidden state are initialized with the input encoded state. Then, at each time-step and for each of the states, the network predicts the difference, or residual, from the previous state to generate the updated states using 2D depthwise separable convolutions (Howard et al., 2017). Depthwise separable convolutions are chosen to preserve the time dimension of the input, which the convolution operates over the two spatial dimensions. The output of the decoder section is the concatenation of the cell states from each time-step (unrolling of the learned time sequence).*

The red portion shown in the figure below corresponds to the additional components required for the Augmented model (described on lines 115-122 of the submitted manuscript)

[Figure]

a) Overall architecture

⊕ Concatenate

b) Decoder

.

Caption: Figure showing (a) the overall network architecture and b) the custom decoder. The red portion refers to the additional components required for the augmented model. The dashed arrows show a process carried out only once (the initialization of the adder). FPN refers to the feature pyramid network, ConvLSTM, the convolutional long short-term memory network, NiN is the network in network module.

Line 92 : What time bin(-s) were used as input (12:00, 00:00, a daily average ?)

Noon samples were used. This information has been added to the last paragraph of the data section.

Line 112 : How does extending to a longer input affects the forecast quality ?

We tested extending to a 5 day input but did not see any improvement in forecast quality. With this longer input the quantity of data to be processed is greater than that for 3 days, which increases the computational expense and data storage requirements. Hence we did not continue with this, or

longer inputs. However, we recognize that in a different domain where other processes are important a longer input may show improvements.

Line 126 : Can you describe these "extensive experimentations" ?

Initially we had used a leave-one-out approach for training and testing of the model. For example, given 30 years of data, train with 29 and test with one, and repeat over all 30 years. However, would result in the use of future data for training, which is not desirable for a forecasting approach. Hence, we moved to the current approach of training initially using ten years, updating the model weights for future time periods. We tested different training periods (10 vs 20) and also different numbers of months to include in training our monthly models. The current configuration led to the best results.

Line 129 : "Chosen to be 10 years". Why is it so ?

The choice of 10 years is a compromise between having enough data to provide the model with representative conditions from which it can learn, and not having the approach become too data-heavy. Because the model already performs reasonably well with this training approach, it could be used alternatively with other data sets for which a shorter time series is available (eg. an AMSRE/AMSR2 unified data set available from 2002, which would have higher spatial resolution, although also different variables).

Line 135 : This processing logic should definitely be represented in a figure.

Good point. We agree and have prepared a figure (below with text from manuscript lines 127-134, modified here)

*For each month of a year a separate model is trained on data from the given month as well as the preceding and following month. For example, the 'April model' is trained using data from March 1 to May 31. This monthly model is initially trained on data from a fixed number of years, chosen to be 10 years. After this initial experiment, to predict each following test year i, using a rolling forecast prediction, the model from year i−1 is retrained with data from year i−2 and also, data from year i−1 is used as validation for early stopping criteria and to evaluate the training performance. For example, if the initial model is trained on 10 years, data from year 11 is used as validation and first predictions are launched at year 12. The model for year 11 is initialized with weights from the 10-year model and retrained with data from year 10, validated on year 11 and predicts year 12. The model for year 12 is then initialized with weights from the year 11 model, retrained with data from year 11 and validated on year 12 to predict year 13. This process is used to produce forecasts of sea ice presence for years 1996 to 2017.*

[Figure]

Line 166 – 167 : The end of the sentence doesn't make sense. Consider reformulating.

This has been changed to "Using additional climate variables for the input of the Augmented model is shown to be beneficial (Fig 1d,e,f). In the periods where the Basic model is worse than climate normal, the Augmented model has better accuracy, for example for the April model at lead days 60-80, and the August model, lead days 75-90..

Line 181 : It seems counterintuitive. Can you explain why ? The reason for the higher Brier score of the Basic model in comparison to augmented here may be because the September model uses training data over August/September and October. We hypothesize that the trend over this period may be less representative of more recent ice conditions than breakup, which may make the additional data used in the Augmented model un-helpful.

Figure 1 : Y-Axes for subfigures d-e-f should be Accuracy differences or ΔAccuracy.

This has been changed.

Line 220 : What would you link the lower accuracy in "central region" ? Is it the higher variability of the freeze-up pattern or to climate variables that are, given the distance to stations, less reliable in such areas ?

We assume the reviewer is referring to the lower performance of Basic model in the central regions relative to the coast in December for a 30 lead day forecast. In this case the degradation was because freeze-up was too late in the model in the central regions. While the climate variables could be less accurate due to their distance to station data (assuming station data are assimilated in a reanalysis and are accurate) our experiments are not set up to evaluate this because we are using ERA5 as our "observations" for comparison.

Lines 236, 246 & 249 + Figures 8 & 9 : From what I know, it should be written Quaqtaq, not Quataq (https://www.makivik.org/quaqtaq/).

Thank you very much. This has been corrected.

**References**

Andrews, J., Babb, D. and Barber, D.G. (2018), "Climate Change and Sea Ice: Shipping in Hudson Bay, Hudson Strait and Foxe Basin (1980-2016)", Elementa, 6,  DOI: 10.1525/elementa.281.

Askenov Y., Popova E.E., Yool, A.,  Nurser, A.J.,  Williams, T.D.,  Bertino, L. and Bergh, J. (2017), On the future navigability of Arctic sea routes: High-resolution projections of the Arctic Ocean and sea ice, Marine Policy, 75, 300-317.

Bushuk, M., Msadek, R. Winton, M. Vecchi, G. A., Gudgel, R., Rosati, A. and Yang, X., (2017), "Skillful regional prediction of Arctic sea ice on seasonal time scales", Geophysical Research Letters, 44, doi: 10.1002/2017GL073155.

Dirkson, A., Merryfield, W.J., and Monahan, A.H., (2019) "Calibrated probabilistic forecasts of Arctic sea ice concentration", Journal of Climate, 32, 1251-1271.

Dirkson, A., et al. (2021) "Development and Calibration of Seasonal Probabilistic Forecasts of Ice-free Dates and Freeze-up Dates." Weather and Forecasting 36.1: 301-324.

Dupont, F., Higginson, S., Bourdallé-Badie, R., Lu, Y., Roy, F., Smith, G. C., Lemieux, J.-F., Garric, G., and Davidson, F.: A high-resolution ocean and sea-ice modelling system for the Arctic and North Atlantic oceans, Geosci. Model Dev., 8, 1577–1594, https://doi.org/10.5194/gmd-8-1577-2015, 2015.

Gignac, C., Bernier, M., & Chokmani, K. (2019). IcePAC–a probabilistic tool to study sea ice spatio-temporal dynamics: application to the Hudson Bay area. The Cryosphere, 13(2), 451-468.

Hochheim, K. P. and Barber, D. G. (2014), An update on the ice climatology of the Hudson Bay system, Arct. Antarct. Alp. Res., 46, 66–83.

Saucier, F., Senneville, S., Prinsenberg, S., Roy, F., Smith, G., Gachon, P., Caya, D., and Laprise, R.: Modelling the sea ice-ocean seasonal cycle in Hudson Bay, Foxe Basin and Hudson Strait, Canada, Climate Dynam., 23, 303–326, 2004.

Sigmond, M., Reader, M.C., Flato, G.M., Merryfield, W.J. and Tivy, A. (2016) "Skillful seasonal forecasts of Arctic sea ice retreat and advance dates in a dynamical forecast system", Geophysical Research Letters, 43, 12,457-12,465, doi:10.1002/2016GL071396.

Zampieri, L., Goessling, H. F., & Jung, T.J. (2018). Bright prospects for Arctic sea ice prediction on subseasonal time scales. *Geophysical Research Letters*, 45, 9731– 9738. https://doi.org/10.1029/2018GL079394

---

## Author Comment (AC2)

**Reply to Reviewer 2- tc-2021-282 Asadi et al. 2021 - Probabilistic Gridded Seasonal Sea Ice Presence Forecasting using Sequence to Sequence Learning**

We sincerely thank the reviewer for the thorough review and excellent comments. Reviewer comments are shown in black, our response are shown in blue.

The authors present a fascinating application of machine learning techniques to better predict ice presence/absence within Hudson Bay, using ERA5 data as an input. The results show promise in helping plan shipping operations around the ice-free season, however, the clarity of these results is lost in lengthy wording. It is recommended that the authors read through the document for grammatical errors and places where the wording of sentences can be made more succinct. This article can become much more impactful and easier to read with more 'straight to the point' sentences.

Thank you for this comment. We agree and will revise the wording throughout the manuscript to make the manuscript more readable.

General comments:

- Ensure you are consistent using 'freeze-up' with a hyphen throughout the document, and choose either 'breakup' or 'break up' to use throughout the document

Thank you. This has been corrected.

- I am aware that it is difficult to phrase sentences when discussing the number of lead days and the two models, however, I found most sentences discussing these topics hard to read. For example, line 149:

'For example, the top row of Fig 1b shows the accuracy of forecasts launched in January using Basic model for forecast lead days of 1 to 90. E.g., the first top-left box in this figure (Fig 1(b)) corresponds to the average accuracy after 1 day forecast for all forecasts launched between January 1 and January 31, ending in January 2 to April 1 and the second box corresponds to average accuracy of forecasts launched between January 1 and January 31 ending in January 3 to April 2.'

I think it would be easier if you use articles when you are referencing lead days or models. For example: 'the Basic model' or 'a 1 day forecast'. This would make your sentences flow better while reading them, which would communicate your results more efficiently.

We agree the wording can be improved and will take this into account by doing a thorough revision.

- The results section has some statements that are more suited towards the discussion section, however I see your discussion and conclusion section are combined. I'm unsure if the section headers are pre-determined by the journal, but if they are not I would suggest making section 6 'Results and Discussion', and section 7 'Conclusion'. This would allow you to discuss your results more in depth as you present them, as I feel like some of your results could be discussed more in depth.

- Throughout the document, you abbreviate some month names and use the full name for others. You should pick one method and stick to it throughout (i.e. always abbreviate or always use the full word).

These have been fixed using the full word except for the figures, which still have the abbreviations.

- There is a comment in the specific comments regarding this, but you should include some discussion regarding the resolution of your results, and how this may impact the use of your results for port-specific operations. I am a little wary of how the land mask may impact how 'close' the pixel you use to represent the port is to the actual port in question. A figure representing this may add some clarity.

We have put together a map of the port locations, including one requested at Sanirajak (formerly known as Hall Beach). The map of the study region is shown below with the port locations shown in red. The Insets show the port location (red) and the nearest point on the model grid (blue) that is outside of the land boundary (where landmask from ERA5 is less than 0.6), in addition to a bounding box that approximates a grid cell. The model grid point near Quaqtaq is located (correctly) in the water region because the landmask from ERA5 has a low value in that region due to the low elevation.

[Figure]

Specific comments:

Line 3 – You may be limited on word count in your abstract, but I think it would be helpful if you stated the type of data you are feeding into your ML system to derive these predictions.

Modified

Line 3 – recommend changing to "Given the recent observations of the declining trend"

Modified.

Line 6 – recommend changing to 'within a 7-day time period', unless you define why a 7-day time period is 'valid' in the manuscript?

Modified.

Line 8 – The introductory sentence needs a little bit of work. I would recommend removing 'northern communities' as you do not speak of them in the rest of the introduction. Maybe focus more on the topic of shipping and why ice forecasting is vital for shipping operations in this introductory sentence. OR add in reference to communities, and why they rely on ice.

Thank you. We have changed to "Sea ice presence is an important variable for shipping operators in the Arctic and surrounding seas as it poses a significant hazard to their operations. For ships with little or no ice-breaking capability, the timing of freeze-up and break-up defines the period over which shipping operations can be carried out. For ships with some ice-breaking capability, the predicted ice cover along a proposed shipping route provides information on transit time and is also required for accurate weather forecasts in ice-covered regions."

Line 12 – Could you expand on what you mean by 'Typical approaches are usually statistical or dynamical in nature.'? Maybe add a reference to examples of these? I see that you go more in depth in the next paragraph into dynamical forecasts, but what about statistical like you mentioned earlier?

We have added the following text to the introduction: Typical approaches are usually statistical or dynamical in nature. Statistical models have included multiple linear regression (Drobot et al., 2006), or Bayesian linear regression (Hovarth et al., 2020), whereas by dynamical approaches we are referring to those that use a forecast model solving the prognostic equations governing evolution of the ice cover (Askenov et. al., 2017, Sigmond et al. 2016). An excellent overview is given in Guemas et al. (2014).

Line 15 – I would recommend splitting this up into two sentences, breaking it up at one of the commas Modified.

Line 16 – remove 'the summer of' before 2008, as you have already indicated that this study was in the spring and summer Modified.

Line 18 – I am not too sure what you mean by 'skill'. Do you mean the forecasts ability to predict ice? There may be a better way to word this to avoid ambiguity. We agree "skill" was not specific enough.

We have changed this to "A comparison between pan-Arctic and regional forecast skill was carried out by (Bushuk et al. 2017), where skill was assessed using the anomaly correlation coefficient (ACC) between sea ice extent derived by applying a threshold to an ensemble-mean sea ice concentration and sea ice concentration from passive microwave data, and detrending both. It was shown that the ACC of seasonal forecasts in specific regions was dependent on the region and forecast month."

Line 20 – It might be nice to list some environmental controlling factors in brackets, like: (i.e. wind speed and direction, tides)

Line 24 – Recommend to change to 'Both of these approaches determine…' Modified

Line 28 – Change to 'composed of sea ice concentration data…'Modified

Line 30 – Remove 'good' Modified

Line 30 – Would help the reader if you included where the mean September sea ice extents from 2017 came from (ice charts? Passive microwave data?)

Changed to "Their predictions were in agreement with the mean September sea ice extent from 2017 where the sea ice extent is the total area in a given region that has at least 15\% of ice cover, according to passive microwave data."

Line 43 – 'calibrated probability of ice': presence or concentration?

Line 64 – Need to define 'SST'  Modified

Line 65 – Doesn't ERA5 have a 31km resolution? I would state this plainly so the reader knows what resolution your results are.

This information had been added to the manuscript (lines 52, 92 and 146)

Line 74 – Would recommend shuffling around this sentence: 'Shipping traffic is also generated by mining, fishing, tourism and research activities, being mostly confined to the ice-free and shoulder season'.

Line 84 – 'In Seq2Seq learning, which has successful applications in machine translation' Modified

Line 87 – Recommend to spell out 'two' Modified

Line 88 – suggest removing 'part' Modified

Line 92 – In line 54 you use the double wavy equal sign, but here you use a single wavy line. I would recommend picking one and keeping it consistent throughout. Modified

Line 94 – Recommend to change to: 'The encoder section of the Basic model takes the last three days of environmental conditions as an input' Modified

Line 97 – Remove 'so as' Modified

Line 99 – May be better to spell out 'LSTM' in full form Modified

Line 101 – Recommend rewording the last sentence for clarity: 'The output to the encoder is a single raster with the same height and width as the input, but a higher number of channels to represent the fully encoded system state.' Modified

Line 115 – Remove 'so as' (try and write sentences as simply as possible, i.e. with as little unnecessary words) Modified

Line 128 – Just verify that your quotation is facing the correct way before 'April' Modified

Line 137 – How did you determine what learning rater and momentum to use? We used the default Keras stochastic gradient descent (SGD) optimizer parameters (learning rate = 1e-2, momentum = 0.9). We also used a learning rate decay of 1e-4 and L2 regularization of 0.0003 with clipnorm=True. These parameters were determined as part of an initial hyperparameter search carried out at the beginning of the study when the compatibility of the data and the model architecture are investigated.

Line 142 – Suggest to remove 'coming', or replace with 'derived' Modified

Line 151 – I would recommend changing the formats of your dates here: 'forecasts launched between 1-31 January, ending in 2 January to 1 April, and the second box corresponds to average accuracy of forecasts launched between 1 – 31 January, ending in 3 January to 2 April. Changed to "forecasts launched between January 1 to 31, ending in January 2 to April 1, and the second box corresponds to average accuracy of forecasts launched between January 1 to 31 ending in January 3 to April 2."

Line 154 – This sentence needs a lot of work: suggest removing 'very' and changing 'on January' to 'of January'. As well, are you indicating that the accuracy is close to 100% for

both January and the span of January – March (this is not clear)? It would be helpful if you stated the actual accuracies.

Thank you. The accuracies are close to 100% for lead days from the beginning of January to the end of March. We will revise the text to include this.

Line 155 – This sentence struggles with the same structural problems as the first, I would recommend rewording to something like: 'In contrast, for forecasts at the beginning of the open water season (June and July), the climate normal struggles to accurately capture the ice cover for lead times of 1 to 50 days likely due to inter-annual variability and the impact of climate change'. You might also want to indicate what climate change has to do with this (i.e. 'lengthening of the open water period due to climate change')

Line 165 – This sentence also needs to be reworded, I have underlined grammatical errors: 'Using additional climate variables for the input of the Augmented model is showing its impact here where in the periods that Basic model is worse than climate normal (Fig 1d), the Augmented model has better accuracy and is closer accuracy to climate normal.

We will use clearer wording in the revised manuscript.'

Line 169 – Double check if it should be 'the climate normal' or 'climate normal' Checked

Line 179 – Spell out 'April' fully, as you have spelled out every other month Modified

Line 202 – 'Observations' should not be capitalized Modified

Figure 4 – Include units for Latitude and Longitude, and capitalize the words in your legend Modified

Figure 5 – Units for lat and long Modified

Line 212 – 'Figure 5 and 6 show the overall…' Modified

Line 215 – 'The freeze-up accuracy maps at Fig 5 show that except the Basic model's prediction at 30 lead day (Fig 6b), other maps are showing similar patterns of accuracy.' This sentence needs reworking – would recommend flipping the sentence, so you are presenting the positive results first, then adding on the Basic model's prediction after. Modified

Line 221 – 'compared' instead of 'comparing' Modified

Line 222 – Capitalize 'fig 6a' Modified

Figure 6 - Units for lat and long Modified

Line 227 – I would recommend changing all of your dates to the format: '1 Oct to 31 Jan'. This is a more standard way of presenting dates and is more simplistic. We have revised the dates.

Line 233 – 'Compared' instead of 'comparing' Modified

Line 234 – Change to 'its accuracy over the breakup season…' Modified

Line 235 – Since you discuss the break up at three sample ports, and present the results in Figures 8 and 9, I think it would be important to include a map of these three locations, indicating which pixels you use to extract this data. I am curious how the land mask affects the data, i.e. how close are the pixels you are using to the actual port? Since you are using 31km ERA5 data, I would suspect that the pixel you chose to represent each port is actually a distance away from the actual dock. In the end, I guess I am a little wary of how applicable your results are to local communities, as they are likely more impacted by ice break up on a smaller scale along the coast (for hunting and travel), whereas shipping operations are more concerned of the large scale ice break up along shipping corridors. Some discussion of how the scale of your results impacts how they are used by different groups may help address this.

We have included a map of the port locations, indicating the locations of the pixels used to extract the data. As the reviewer has pointed out these locations are a distance from the actual port. For this reason, and also because i) atmospheric conditions represented from ERA5 would be different from those at the actual port locations ii) we do not have a complete description of sea ice conditions that can represent the complexity of port conditions, our model output is expected to be more representative of offshore conditions, which is important for route planning. We will revise the manuscript wording to reflect this.

Line 237 – Capitalize 'figures' Modified

Line 242 – Would recommend moving the figure reference to the end of the sentence, and putting it in brackets OR starting the sentence with 'In figure 8, 30 lead day predictions for freeze-up are more…' Modified

Line 242 – Any idea why this is? I am curious why the predictions varied at the different town ports and would think a discussion of this would add to your paper. We agree the predictions for the various ports is interesting. It should be noted that the range of dates covered in the x-axis (Observed dates) varies between the ports. For freeze-up, the narrowest range is for Churchill (approximately 1 month), whereas Quataq and Inukjuak both have a wider range (approximately 6 weeks). We will look into this by comparing the variability (over the years) of sea ice break-up and freeze-up dates at the various locations

with the variability (over the years) of the input data. The sea ice break-up and freeze-up used in the comparison is from the observations (ERA5, based initially on passive microwave data) and may have outliers not captured by our train-test procedure. We are also planning to compare the predictions at the various ports with data from regional ice charts provided by the Canadian Ice Service. The regional charts are weekly analyses of ice cover (concentration and stage of development of the ice) that are based on manual interpretation of synthetic aperture radar (SAR) imagery in addition to other sources, such as passive microwave and visible imagery and ship reports. A similar comparison has been done in Gignac et al. (2019).

Line 252 – If you have space in your word count, I would recommend listing the 8 variables used in the Basic model, and the other variables added to the Augmented. This would help refresh the reader's memory as to how these two models vary. Good idea. Thank you

Figures 8 and 9 – If possible, the font size should be increased, particularly for your axis labels. This might take some reorganizing of your figure boxes – maybe you could rotate the 'model' and 'day' labels on the far left of your figures? Modified

Askenov Y., Popova E.E., Yool, A., Nurser, A.J., Williams, T.D., Bertino, L. and Bergh, J. (2017), On the future navigability of Arctic sea routes: High-resolution projections of the Arctic Ocean and sea ice, Marine Policy, 75, 300-317.

Bushuk, M., Msadek, R. Winton, M. Vecchi, G. A., Gudgel, R., Rosati, A. and Yang, X., (2017), "Skillful regional prediction of Arctic sea ice on seasonal time scales", Geophysical Research Letters, 44, doi: 10.1002/2017GL073155.

Dirkson, A., Merryfield, W.J., and Monahan, A.H., (2019) "Calibrated probabilistic forecasts of Arctic sea ice concentration", Journal of Climate, 32, 1251-1271.

Dirkson, A., et al. (2021) "Development and Calibration of Seasonal Probabilistic Forecasts of Ice-free Dates and Freeze-up Dates." Weather and Forecasting 36.1: 301-324.

Drobot, S.D., Maslanik, J.A., and Fowler, C., (2006) "A long-range forecast of Arctic summer sea-ice minimum extent", Geophysical Research Letter, 33, L10501, doi:10.1029/2006GL026216.

Gignac, C., Bernier, M., & Chokmani, K. (2019). IcePAC–a probabilistic tool to study sea ice spatio-temporal dynamics: application to the Hudson Bay area. The Cryosphere, 13(2), 451-468.

Guemas et. al., (2014), "A review on Arctic sea ice predictability and prediction on seasonal-to-decadal time scales", Quarterly Journal of the Royal Meteorological Society, doi:10.1002/qj.2401.

Horvath, S. et al., (2020) "A Bayesian logistic regression for probabilistic forecasts of the minimum September Arctic sea ice cover", Earth and Space Science, 7, doi:10.1029/2020EA001176.

Sigmond, M., Reader, M.C., Flato, G.M., Merryfield, W.J. and Tivy, A. (2016) "Skillful seasonal forecasts of Arctic sea ice retreat and advance dates in a dynamical forecast system", Geophysical Research Letters, 43, 12,457-12,465, doi:10.1002/2016GL071396.

---

## Author Comment (AC3)

**Reply to Reviewer 3- tc-2021-282 Asadi et al. 2021 - Probabilistic Gridded Seasonal Sea Ice Presence Forecasting using Sequence to Sequence Learning**

We sincerely thank the reviewer for the thorough review and excellent comments. We have mainly provided responses and clarifications for the detailed questions, not addressing the typos and minor comments directly, but these would be corrected in the revised manuscript and will greatly improve the quality of the manuscript. Reviewer comments are shown in black, our responses are shown in blue.

The authors present a new approach for forecasting sea ice presence in the Hudson Bay area using machine learning techniques. The study presents models which use the Sequence-to-Sequence Learning framework to predict probabilities of sea ice presence for up to 90 days lead time. The authors suggest two somewhat different models, which are applied in hindcasting experiments, where they exhibit slightly more skill than "climate normal"-predictions especially in the breakup season. The models are also evaluated for their ability to predict freeze-up dates and breakup dates.

The study has a clear motivation, is well structured, and applies new (as to my knowledge) methods in a promising way. The text is short and precise. The general setup of the experiments is described well, however, as I'm not an expert in ML, I cannot judge the parts of the paper that go into technical details of the ML process. The results are presented clearly, but I miss a broader discussion of the results and which conclusions can be drawn from them. Especially as the motivation of the study is to develop new methods to support maritime users with new operational forecast products, a comparison to existing products would be valuable. Also a short assessment/discussion of the applicability to operational forecasting is missing in my opinion.

Please find here some general comments, followed by comments related to specific line numbers. Text in quotation marks after the given line number refers to the original text of the manuscript. Text in quotation marks in the following line is my suggestion of how to replace the original text.

%%%%%%%%%%%%%%%%%%%%

General comments

%%%%%%%%%%%%%%%%%%%%

A) The captions of Figures 1-3 do not only explain the figure but also contain statements about the shown results. This is a new approach to me. If the journal allows for that, I would not object, but I don't think it is common style.

We did this to help the reader interpret the figures more readily. We are not aware of limitations on figure captions for this journal.

B) For Figure 4 you compare predicted ice presence with observed ice presence. The observations are calculated from SIC from ERA5 by using a threshold of 15 %, while the ice presence from the forecasted probabilities is calculated with a threshold of 50 %. As the model is based and trained on

SIC from ERA5, why do you use a threshold of 50 % and not of 15 %? Or at least the same threshold for both?

While the model is trained on SIC from ERA5, other variables are also used, and it is an ice presence probability that is the model output. A probability over a grid cell is different from an ice concentration in that it indicates the probability of an event, which here is that the SIC is greater than 15%. A threshold of 15% is chosen for SIC, which is a common value used in the sea ice community (Stroeve 2015, Gignac et al, 2019). Note that the same thresholds were used in Andersson et al. (2021) in their study of sea ice prediction using a related convolutional neural network approach.

C) I understand that the motivation of your study is (at least partly and in the long run) to improve operational forecasting of sea ice conditions in the Hudson Bay area. In line 254-257 you explain the technical advantages of the ML approach compared to standard numerical models (=reduced computational costs). The paper would benefit from also looking into the results/skill of the ML models compared to standard models. Is your approach not only faster but also better than currently used forecasting systems? Or is it so much faster that it is useful despite of a possibly lower quality? Or is it worse/better only for some lead times? It would be interesting to see how your model compares e.g. to the S2S-forecast of ECMWF (up to 60 day forecast at 1/4 degree resolution).

https://www.ecmwf.int/en/forecasts/dataset/sub-seasonal-seasonal-prediction https://apps.ecmwf.int/datasets/data/s2s-realtime-daily-averaged-ecmf/levtype=sfc/type=cf/

Comparing your models to the climate normal is a very good and valid first step. The comparison to a numerical forecast would however be a very interesting addition.

We agree a comparison to other data, such as the subseasonal to seasonal predictions available from ECMWF, would be a good addition to the manuscript. The sea ice information at the link provided (to the S2S ECMWF, Realtime Daily Averaged data) is available for forecast lead days up to day 46, with output twice a week, over the years 2015-2021. We find the spatial resolution to be 1.5 degrees x 1.5 degrees. We are not certain therefore if this is the precise data set the reviewer is referring to. We note Zampieri et al (2018) carry out an extensive comparison of various S2S sea ice forecasts for the Arctic, one of which is from the ECMWF system. In Zampieri et al. (2018) it is stated "The sea ice concentration fields from the S2S database are provided on a 1.5° × 1.5° longitude-latitude grid, although the sea ice models run are at higher resolution (from 0.25° to 1°)".

At this stage, we have carried out a comparison for the year of 2017 between our model forecasts and those from the S2S system. To do this the same thresholds are applied to both the predicted sea ice presence (a probability greater than 50% corresponds to ice) and the sea ice concentration from ECMWF (a sea ice concentration greater than 15% corresponds to ice). The ECMWF predictions are launched twice a week (Monday and Thursday). For the comparison, the data from our system (CNN-LSTM model predictions, climate normal, and observations) are extracted for the same launch dates as those used by ECMWF. The ECMWF data was interpolated to our 31 km grid resolution using a nearest neighbor approach. The accuracy is calculated in the same way as for the other accuracy plots in the submitted manuscript, through addition of the true positives for ice and true positives for water divided by the total number of points considered. For example, to evaluate the Basic model true positive occurs when both the predicted probability is greater than 0.50 and the observed sea ice concentration (here from ERA5) is over 0.15, while a true negative occurs when both the predicted probability is less than 0.50 and the observed sea ice concentration is less than 0.15. The total number of points is the total number of non-land points times the number of days considered in the accuracy calculation. For each monthly model at each lead time, the number of days corresponds to the number of days in the given month (egl, 31 days for January).

Results are shown for each month of 2017 in the figure below for the Basic model. The accuracies noted for the Basic model and Climate normal are different than those given in Fig 1 of our submitted manuscript because the statistics are only calculated over non-land grid cells. Due to the spatial resolution of the ECMWF data, there are only a few points in Hudson Strait and Foxe Basin. It can be seen in the figure below that during break-up (May, June and July) the proposed method has a higher accuracy than the ECMWF forecasts, whereas during freeze-up (in particular November) the ECMWF forecasts have a much higher accuracy than those from the proposed approach. The poor performance of Basic model in November is due to the opening of the Kivalliq polynya in the north western portion of the domain. This is a large latent heat polynya that is sustained in part due to strong offshore winds. The Basic model predicts freeze-up too quickly in this region, in comparison to the observations, which here are the sea ice concentration from ERA5. The ECMWF model performs better, likely because it has a sea ice model coupled to the atmosphere. When ice starts to form this reduces the heat exchange from the ocean to the atmosphere, and the rate of ice growth slows. Our approach may not have trouble representing this phenomena because the patterns learned associated with the variables used for training could correspond to ice cover. We will need to look into this in more detail, bringing in passive microwave data (Bruneau et al. 2021), keeping in mind this ERA5 sea ice concentration is based on passive microwave data, with values less than 0.15 truncated to 0.

We are currently processing other years over which the S2S forecasts are available. Note that the observations here, that are used in the accuracy calculation, are from ERA5 (consistent with what is done in the manuscript). This is relevant because our model is trained with ERA5.

[Figure]

Caption: Accuracy of sea ice presence for each month of 2017 for forecasts up to 46 days. The ECMWF data are from the subseasonal-to seasonal database (s2s-realtime-daily-averaged). The accuracies shown here for the Basic model and Climate normal are different than those in Fig 1 because the points considered correspond to the landmask of ECMWF, which has a coarse grid (1.5 degrees x 1.5 degrees) and therefore some regions, such as Hudson Strait, and not well represented, and the data used correspond mostly to Hudson Bay.

D) With the Basic and Augmented models you introduce two approaches, which you compare with each other throughout the paper. However, I don't find a conclusion/discussion about which of the two models you would suggest in the end. Is it worth the effort of the Augmented model, which needs more input data, or is the Basic model sufficient for the purpose? Or are both needed, for different purposes? Maybe you can here also explain/speculate why the Augmented model is considerably worse than the Basic model in Figure 2c. [This text could extend the summary given in lines 258-265.]

Figure 2c shows the Brier score difference for the two models. A Brier score of zero indicates an optimal result. With this in mind, the negative values in this plot indicate the Augmented value is better than the Basic model for most of the ice season. The Augmented model is worse for the September forecasts (60-90 days, or sea ice states from Nov 1 to Dec 30), and the October forecasts (30-40 days, or sea ice states from Nov 1 to Dec 9). This suggests the patterns between the input features and ice presence by the Augmented model are not able to represent the sea ice evolution in November. Outside of these dates there is a small improvement when the Augmented model is used. Fig 7 shows time series of the Basic and Augmented model, and climate normal, over

1996-2017. The trend lines indicate the accuracy of freeze up and break up is slightly improved with the Augmented model, in comparison to the Basic model.

E) For your hindcasts you use input data from ERA5, which is a reanalysis product that is not available in real time. Hence, when one wants to apply your method for forecasting *future* conditions, other input data need to be used. It would be good to elaborate on this topic in a paragraph in the Discussion or Outlook sections. Is it difficult/problematic to switch to other input data? Can the trained monthly models be applied if the forecast is started based on other data for the 3 historical days? And/Or what else is still needed before your models can be used for operational forecasting? This would be a good topic for an Outlook-section/paragraph.

Thank you for this comment. We do not envision it to be difficult to switch to other input data from the point of view of model architecture. However, the presence or absence of sea ice at different locations will be dependent on different driving factors, hence in other variables (such as sea ice thickness) may be critical. In addition, because this model is trained using ERA5 it will learn the dependencies and patterns in these data. We recommend that if one was to use input data from a different reanalysis that they fine-tune the existing weights to account for the different data dependencies in the input data (in particular consider that only a subset of model variables are used, dependences present in one subset may be partially considered in a different subset for a different model). Finally, while the model here is demonstrated in hindcasting mode, it can (and is intended) to be used in forecast mode. In forecast mode, given an input time series of three days, forecasts can be generated up to 90 days lead time.

%%%%%%%%%%%%%%%%

Specific comments

%%%%%%%%%%%%%%%%

Title
##############
In the introduction you mention that the novelty is that your forecast is "spatiotemporal". Hence I wonder why you don't use this word in the title.
Good point – thank you

Abstract
##############
2-3 Would ML approaches be less important without global warming? Suggestion: Remove "Given ... global warming". Good point – thank you

"a daily spatial map" Isn't the clue of your study that you provide several "daily spatial mapS", namely 90 for a 90-day forecast?Good point – thank you

Introduction

##################

9, 10, 12, 14, 82 Be more clear on the terms "short-term", "longer term", "seasonal" and "medium-term" forecasting. Thank you for pointing this out. We have corrected the text and added more context. "Sea ice forecasting needs to be carried out at various spatial and temporal scales to address different requirements of stakeholders. Short-term forecasts (1-7 days) at high spatial resolution are important for day-to-day operations and weather forecasting (Carrieres et al., 2017), whereas longer term (eg. 60-90 day) forecasts are desired by shipping companies and offshore operators in the Arctic for strategic planning (Melia et al. 2016). In this study we are interested in these longer term forecasting methods, which we will refer to as seasonal forecasting."

Maybe mention the lead-time used in Zhang et al. (2008) We have added text to describe this study. "This study used a coupled ice-ocean model forced by a year of atmospheric forcing data taken from a representative ensemble."

"governing" Do the equations govern the physics? I'd suggest "describing"

"This is a key advantage of..." Suggestion: "This disadvantage can be overcome by using..."

"to perform" "for"

"Their results are" "The results were"

29, 31 In line 29 you write that the model predicts sea ice concentration but in line 31 you present results for ice extent. I'm not sure if one can assume that everyone knows the relation between SIC and sea ice extent. Thank you. We will define this difference in the revision.

"September sea ice minimum." minimum extent? minimum thickness? Modified to indicate sea ice extent

"This study" This is misleading because it could mean your own study. Better use "They" or "Hovath et al. (2020)". modified

"was found the uncertainty" "was found that the uncertainty" Modified

"that is closer to what is proposed here" As we don't know yet, what you will propose, this information is not very useful here. Good point, we removed this statement.

"probability of ice at" "probability of ice presence at" modified

Data

##################

"data from 1985-2017 is" "data from 1985-2017 are"

"the following input variables" "the following 8 input variables". This helps explaining the number 8 in line 96.

"V-Component" "V-component"

replace "and" before "landmask" by comma

Study region

######################

Here would be a good place for a map which also indicates the ports used later on.

Remove the parenthesis if Foxe Basin can be shown in a map.

We agree with the above two points and will be adding a map in the revised manuscript. We have put together a map of the port locations, including one requested at Sanirajak (formerly known as Hall Beach). The map of the study region is shown below with the port locations shown in red. The Insets show the port location (red) and the nearest point on the model grid (blue) that is outside of the land boundary (where landmask from ERA5 is less than 0.6), in addition to a bounding box that approximates a grid cell. The model grid point near Quaqtaq is located (correctly) in the water region because the landmask from ERA5 has a low value in that region due to the low elevation.

[Figure]

"Recent decades"

For me, 1985-2017 includes several "recent decades" and one could get the impression that lines 77-79 are not valid for "recent decades". So maybe consider re-phrasing "recent decades" to "In recent years" or using the term "trend"?

Forecast model architecture

####################

85/86 "sequence of inputs"/"sequence of outputs": It would be helpful if you could mention (maybe in a new sentence) some examples for "input" and "output" for the application in this study. I guess input includes SST, t2m, winds, etc. and output is ice presence probability? Yes, that is correct, We have modified the text to reflect this."The encoder component transforms a given input (here, a set of geophysical variables such as sea ice concentration, air temperature etc.) to an encoded state of fixed shape, while the decoder takes that encoded state and generates an output sequence, here, a sea ice presence probability), with the desired length, which corresponds to the number of days of the forecast (90 days)."

"consist" "consists"

Does "desired length" in your application mean number of variables or number of grid cells or number of forecasted days? It is good with a general explanation of the Seq2Seq method like you do here, but for someone not from the ML field, it would also be nice to directly get examples about how the method can be understood for the application of sea ice forecasting. In this study the desired length is the number of forecasted days, which is 90.

89-90  I first understood this sentence such that the encoder part would be called Basic model and the decoder part would be the Augmented model. Can you phrase it differently to make it more clear also for non ML-experts?

We have changed this to "using the encoder-decoder architecture described above, two spatialtemporal sequence-to-sequence prediction models are developed. These will be referred to as the "Basic Model" and "Augmented Model" and are described in Sections 4.1 and 4.2 respectively". We are also adding a figure of the architecture of the model used, which should help clarify this (please see below)

"three days of environmental conditions" Shouldn't it read "environmental conditions of the last three days"? Yes thank you.

Maybe explain why you call the number of input variables "C":

"and C is the number of channels, in this case the total number of input variables (here 8)."

"sequence of extracted feature grid"

Are the feature grids what was called "environmental patterns" in the previous sentence? If so, could you use the same term? If not, could you explain how to get from one to another?

The feature grids are the "environmental patterns" referred to in the previous sentence. We will use this terminology in the revision consistently

"the sequence are" "the sequence is"

97-113 As I don't have a background in AI/ML, I unfortunately don't understand the setup of the model in detail. However, as I can follow the general concept, e.g. what is input and what is output, I think it is OK to keep the text as it is if the targeted audience is AI/ML experts more than sea ice modellers.

We will be adding more description and figures to the manuscript. The other reviewers have suggested this as well. We have two figures in mind, one describing the method in terms of data preparation and train test sequence, and another that describes the model architecture in more detail. Preliminary figures are shown below.

The first figure shows the train/test sequence. It is used to describe the following text from the manuscript,

*For each month of a year a separate model is trained on data from the given month as well as the preceding and following month. For example, the 'April model' is trained using data from March 1 to May 31. This monthly model is initially trained on data from a fixed number of years, chosen to be 10 years. After this initial experiment, to predict each following test year i, using a rolling forecast prediction, the model from year i−1 is retrained with data from year i−2 and also, data from year i−1 is used as validation for early stopping criteria and to evaluate the training performance. For example, if the initial model is trained on 10 years, data from year 11 is used as validation and first predictions are launched at year 12. The model for year 11 is initialized with weights from the 10-year model and retrained with data from year 10, validated on year 11 and predicts year 12. The model for year 12 is then initialized with weights from the year 11 model, retrained with data from year 11 and validated on year 12 to predict year 13. This process is used to produce forecasts of sea ice presence for years 1996 to 2017.*

[Figure]

The second figure is also shown below. The upper panel shows the overall architecture(described on lines 98-105 of the submitted manuscript, modified here).

*The overall architecture is shown in the figure below (panel a). The encoder starts by passing each daily sample through a feature pyramid network (Lin et al., 2017) so as to detect environmental patterns at both the local and large scales. Next, the sequence of feature grids extracted from the feature pyramid network are further processed through a convolutional LSTM layer (ConvLSTM) (Hochreiter and Schmidhuber, 1997; Xingjian et al., 2015), returning the last output state. This layer learns a single grid representation of the time series that also preserves spatial locality. Finally, the most recent day of historic input data is concatenated with the ConvLSTM output. The encoder provides as output a single raster with the same height and width as the stack of raster data input to the network, but with a higher number of channels such as to represent the fully encoded system state. The final encoded state is then fed to a custom recurrent neural network (RNN) decoder that extrapolates the state across the specified number of time-steps. It takes as input the encoded state with multiple channels and as output produces a state with the same height and width as the input over the desired number of time-steps in the forecast (here 90 days). Finally, a time-distributed network-in-network (Lin et al., 2013) structure is employed to apply a 1D convolution on each time-step prediction to keep the spatial grid size the same but reduce the number of channels to one, representing the daily probabilities of ice presence over the forecast period (e.g. up to 90 days).*

The lower panel shows the decoder (described on lines 106-113 of the submitted manuscript, modified here)

*The custom RNN decoder, as is common of many RNN layers, maintains both a cell state and a hidden state (Yu et al.,2019). First, the initial cell state and hidden state are initialized with the input encoded state. Then, at each time-step and for each of the states, the network predicts the difference, or residual, from the previous state to generate the updated states using 2D depthwise separable convolutions (Howard et al., 2017). Depthwise separable convolutions are chosen to preserve the time dimension of the input, which the convolution operates over the two spatial dimensions. The output of the decoder section is the concatenation of the cell states from each time-step (unrolling of the learned time sequence).*

The red portion shown in the figure below corresponds to the additional components required for the Augmented model (described on lines 115-122 of the submitted manuscript)

[Figure]

a) Overall architecture b) Decoder

I miss a sentence about why you suggest an additional model. What is the (expected) problem with the Basic model or which benefits do you expect from the Augmented model?

The Augmented model was not developed to address a specific problem with the Basic model. It was done to enforce the climate normal, which can help the model generalize, meaning produce better forecasts over a wider range of conditions.

"(e.g., 60 or 90 days)" remove comma

"t2m, u10 and v10"

Why do you use exactly these variables? I could imagine that climate normals of e.g. sea ice concentration or sea surface temperature could also be beneficial to correctly predict sea ice presence.

These variables were chosen because of their availability in both historical data set, and real time (for this application, through the Meteorological Service of Canada GeoMet platform). Since this branch of the network 'augments' the core model, it was desired to keep this flexibility for future development as our computing infrastructure is designed to connect with GeoMet.

Description of Experiments

############################

"required" "requires"

"assess" "assesses"

"the model from year i-1" "the model for year i-1"

End the sentence after "i-2" and start a new one.

and 140 Are "ML models" (line 136) and "neural network model" (line 140) something different? If not, use the same word.

"3 months of year" "3 months of each year"

"thresholded at 15%" This is unclear to me. Do you apply the threshold to convert to ice presence? Yes, that is why we apply the threshold.

131, 132, 142 In line 142 you talk about "test procedure". Is this the same as "validation" mentioned above? Validation refers to the use of the year following those used for training to check early stopping criteria. The test procedure is the same as the prediction procedure, referred to on line 133.

6.1 Presence of Ice Forecasts

##########################

"6.1 Presence of Ice Forecasts"

Shouldn't it be "Forecasts of Ice Presence"?

"test set" What is this? I don't think you have introduced this term before. The test set is the set of days over which the 90 day predictions are launched.

How do you calculate accuracy from the binary forecast map? Accuracy is the ratio of the number of correctly classified pixels (ice or water) to the total number of pixels under consideration. For example, if we consider the accuracy of the Basic model, it is calculated as: accuracy=(TP+TN)/N, where TP is number of true positives (observation and Basic model are both ice), TN is the number of true negatives (observation and Basic model are both water) and N is the total number of points considered, which is the number of non-land points multiplied by the number of days the score is calculated over.

148, 149, 150 etc.I would put a period after the abbreviated "Fig" -> "Fig."

"in this figure (Fig 1(b))" "in Fig. 1b"

"the first top-left" "the first (top-left)

"after 1 day forecast" "after a 1-day forecast"

152-153 Why April 1 and April 2? Wouldn't a 1-day forecast started on January 31 end on February 1, and a 2-day forecast on February 2? Yes, a 1-day forecast started on January 31 will end on February 1. This wording pertained to a 90 day forecast, which would end April 1 if launched on January 1. We will revise the wording to clarify.

"month on January" "month of January"

"consistently" "constantly"

Mention the sub-figure number you are talking about.

"Fig 1d" "Fig. 1d and 1e"

"significantly"

Did you do a statistical test whether it is significant? Otherwise maybe remove the word. We did not do a statistical test and will change the wording.

"early lead times" "short lead times"

"Climate normal" Stay consistent with capital C or not.

163-164 Comparing the Augmented model with the climate normal (Fig. 1e), I don't see an improvement for March/April. I see the Augmented model is better than the Basic model, but actually it is just 'less bad' compared to the climate normal. The accuracy of the Augmented model is not higher than of the climate normal. (You explain this later, so maybe make clear that this sentence only deals with Fig. 1f and not with Fig. 1e.

Thank you for noting this. We will revise the wording

"(Fig 1d)" "(dark areas in Fig. 1d)"

Remove "accuracy" at the beginning of the line.

"90 lead day" "90 lead days"

The "For example..."-sentence is not complete, there is no verb.

Consider to start a new paragraph for the Brier score.

(related to comment for line 148): What is "probabilistic accuracy" compared to "accuracy"?

The term probabilistic accuracy was used to refer to the Brier score, while accuracy is the accuracy calculated using the ratio of true positives and true negatives to the total number of samples. We will put in an equation and revise the wording.

Remove "Also"

"Pt is the model prediction"  Maybe add "... of ice presence probability"

"represents" "presents"/"shows"

"The pattern observed" "observed can easily be mixed with observations, so maybe say "The resulting pattern"

End sentence after "both models" and start a new one for the differences.

"(2c)" "(Fig. 2c)"

"longer lead days" "longer lead times"

"For early lead days" "For short lead times"

"lead day" "lead days" (make sure to do it consistently throughout the paper, e.g. line 211 and in caption of Figure 3)

184-187 I find this sentence is too long. Also, there is inconsistency of the used terms "forecasted probability" and "forecasted probabilities".

When first reading the sentence, it sounds like monthly averaging would make it impossible to provide information on a map. Maybe you can clarify it with "Monthly averaged and domain-integrated accuracy values ..."

194-195 To simplify the explanation of the different dates, I suggest to add the dates in the caption of the subfigures 4a-4c, e.g. "(a) 5 June 2014 (after 30 days)"

"given data" "given date"

"and and" "and the"

200-201 I don't see in the figures that the Basic model would have "increased ice presence probability in the northern part of the domain". If it is important, highlight the area in the plots.

"Observations" "observations"

201-202 "the agreement ... to be in good agreement" Too many agreements.

Figure 1 and 2:

I would suggest to use a diverging colormap when differences are displayed. This makes it easy to see where zero is located and it makes it more clear which plots display differences and which plots display absolute values.This is a good idea. Thank you.

Caption of Figure 1: "Model performance and improvements": Why not call it "Accuracy"? =

Caption of Figure 2: "Brier score of the Basic model (a) and the Augmented model (b) as a function of lead time. Their score difference is shown in (c). Most differences are observed in breakup and freeze-up seasons."

Figure 3: - It would be nice if the aspect ratio of x and y axis was 1, so that the dashed line would be at 45 degree.

- for the text in the legend I suggest "xx lead days" instead of "xx Lead Day"

Caption of Figure 3:

- Would be nice to remind the reader (especially those who only look at the figures and don't read the text) that you are talking about ice presence probabilities/frequencies.

Figure 4: - In order to compare probability maps with ice presence maps it would be nice (if possible) if the plots would have the same size, i.e. smaller plots for those which have no colorbar.

- I would prefer if the height of the colorbar was the same as the height of the map-plot.

- Make sure to use "Climate normal" or "Climate Normal" consistently.

Caption of Figure 4:

- The figure does not illustrate "the May models" but the forecasted conditions. Hence a suggestion for rephrasing: "Ice/water distribution in the model domain as observed and forecasted by Basic and

Augmented models for a forecast started on 6 May 2014 and lasting for 30 (a), 50 (b), and 70 (c) days, respectively."

6.2 Assessment of operational capability

##############################

"15 continuous days in a row" Either "continuous" or "in a row" is enough.

It can be a bit confusing that "accuracy" here is related to freeze-up/breakup while the same word is used in section 6.1 for ice presence. (Well done in line 215.)

"...prediction is correct." The reader has to infer that 'correct' is translated to 1 and 'not correct' is translated to 0. Yes, this is confusing. We will clarify this.

215-216 Check the grammar of the sentence.

219-212 It is surprising to me that a model can have that much more skill on a 30 day longer lead time. Could you elaborate on possible reasons and/or why the Augmented model is doing a better job? (This should probably go to the Discussion section)

We assume you are referring to Fig 5, where panels b) and c) show the accuracy of freeze up for 30 and 60 days using the Basic model, and panels d) and e) show the accuracy of freeze up for 30 and 60 days using the Augmented model.

Freeze-up dates are checked from Oct 1 to Jan 31.

30-day forecasts would have been launched Sep 1 to Dec 31. These models would have been trained on data from Aug 1 - Oct 31 (for the September model) and Nov 1 to Jan 31 (for the Dec model).

60-day forecasts would have been launched Aug 1 to Nov 30. These models would have been trained on data from July 1 - Sep 31 (for the Aug model) and Oct 1 to Dec 31 (for the Nov model).

The 60 day forecasts may be better than the 30 day forecasts because the air temperature can have more of an impact for 60 day forecasts as the open water season is considered more heavily in the training data for the 60 day model (training data extends into July). Klaus et al (2014) note a dependence of sea ice extent on fall air temperatures during freeze-up in this region. Note for the Basic model the accuracy is quite high along the west coast for 30 days, which could be due to the role of wind in this region, which would be more highly correlated with freeze-up at short time scales.

"breakup prediction ability" Why not call it "breakup accuracy" in analogy to line 215?

"are presented" "is presented"

"fig 6a" Fig. 6a

"variability" Would "interannual variability" be more clear?

"models' accuracy" As climate normal is not really a model, I would remove the word "models'".

"represented" "presented"

"For each prediction... same color." Suggestion: "The respective trends are shown by dashed lines."

229,233 "freeze-up season accuracy" "season" is not necessary.

"both lead days" "both lead times"

"breakup plots (...)" "breakup accuracy (Fig. 7c and 7d)

Move "accuracy" directly after "2%"?

232-233 Is this because the Augmented model uses climate information as input data and hence tends to predict more similarly to the climate normal than the (more independent) Basic model does?

We see more variability in model predictions for freeze-up than breakup in general, and in particular less improvement of the Augmented model in this season. This may be because climatological trends in freeze up have changed in recent years more than those for breakup. We will look into this further.

"is different." "is different (i.e. different x and y axes)."

Refer here to the (new) overview map for locations of the ports.

"both models both lead days" "both models and both lead times"

"as freeze-up for breakup" "for breakup as for freeze-up"

Figures 5 and 6:- Add a space between (a), (b), ... and the subfigure caption text.

- Why did you choose to use a diverging colormap even though the plot does not display differences?

- Remove the grid lines which are drawn around each grid cell in order to make the plot less busy.

Caption of figure 6:- End with a period.

Figure 7:- Add a space between (a), (b), ... and the subfigure caption text.

- You could specify "Freeze-up accuracy" and "Breakup accuracy" also in the y-labels.

- Red and green lines are probably difficult to distinguish for color-blind persons. What about using black for the climate normal and e.g. red and cyan for the two models? We agree this should be modified

Caption of figure 7: "Dashed lines" instead of "Dotted lines"

Figures 8 and 9:- The little red arrows next to the dots are hard to see and the written year numbers don't allow for getting a quick overview of the distribution of the year (e.g. whether the skill is getting better or worse over time). Did you try to plot the dots using a colormap which represents the different years (i.e. colorful dots, and the "valid area" as gray)? Then you don't need the text labels anymore. This is a good idea

- "freeze-up" instead of "Freeze-up" in x-labels and y-labels.

Caption of figure 8 and 9:- Mention that each dot represents one year.

Discussion

###############

In my opinion you should mention somewhere that your model is not (yet) used for forecasting future condition but rather for hindcasting. Thank you. We will clarify that while the model can produce 90 day forecasts, it is evaluated here as in hindcasting mode to demonstrate the concept

"where it takes" "which takes"

"Augmented model showing better scores comparing to" "Augmented model shows better scores compared to"

"analysis on" "analysis of"

"less disperse" less disperse than what? We will clarify that we mean the dates for freeze-up are less disperse than those for break up.

"to accepted" "to the accepted"

References

#################

"shipping" "Shipping"

293, 305, 308 Why do you cite preprints of papers that are several years old?

"S., R., G.," The co-author is called "Graversen R"

"high resolution" "high-resolution"

298, 322 missing volume/issue/page number

"W.-c." "W.-C."?

Additional References:

Bruneau et al., (2021), The ice factory of Hudson Bay: Spatiotemporal variability of the Kivalliq Polynya, Elem Sci Anthro,  9(1), doi:10.1525/elementa.2020.00168.

Carrieres, T. et al., (2017), Sea ice analysis and forecasting, Cambridge University Press.

Melia, N., Haines, K and Hawkins, E.(2016), Sea ice decline and 21st century trans-Arctic shipping routes, Geophysical Research Letters, 43(18), p. 8720-9728.

Stroeve, J., E. Blanchard-Wrigglesworth, V. Guemas, S. Howell, F. Massonnet, and S. Tietsche (2015), Improving predictions of Arctic sea ice extent, *Eos, 96,* doi:10.1029/2015EO031431

Zampieri, L., Goessling, H.F. and Jung, T.J. (2018), Bright prospects for Arctic sea ice prediction on subseasonal time scales, Geophysical Research Letters, 45(18), p. 9731-9738.

---

## Referee Report (RR1)

**"Probabilistic Spatio-temporal Seasonal Sea Ice Presence Forecasting using Sequence to Sequence Learning"**

The manuscript has improved considerably, both in terms of presentation of text, methods (model description, experiment description, skill scores), results, figures and due to new content. The discussion section was extended and includes good new aspects, and additional sections were added in which the proposed Seq2Seq models are compared to ice charts and S2S forecasts. This puts the proposed method in context to other commonly used data products and is a valuable addition.

Unfortunately, I find the new sections (7.2.3, 8, 9, and partly 10) more difficult to read than other sections. First of all, there are many little mistakes like typos, double words, inconsistencies, missing references, wrong figure numbers etc.. I've tried to point some (not all) of these issues out in the list below, but I think these sections need to be revised carefully by the authors. These sections would benefit not only from fixing the little mistakes, but also from an additional attempt to improve readability of the text. The other sections were easier to read for me.

Regarding the revised documents, I noticed that the document with tracked-changes does not match the revised manuscript (without tracked-changes). From the properties of the pdf files I could see that the tracked-change-document was older, hence I did this review based on the non-tracked-change document and line numbers are referring to this. Nevertheless, I would wish that I would not need to spend time to check for such inconsistencies.

I'm also confused to see that some of my earlier proposed corrections have apparently not been processed (especially regarding typos in the Reference section.) It's completely OK to not implement my *suggestions* if you don't like them. But I couldn't find an indication in the author-reply document whether you have overlooked the proposed *corrections* or whether you just intend to implement them sometime later. Therefore, I refer to them again.

General
#################

A) Good that you found the inconsistency of the ERA5 landmask and the landmask used for SIC.

B) In line 78 you mention that you use hourly data at 12:00 for each day. Was it a deliberate decision to not use daily averages? I would assume that the models could maybe do a better job if they used daily averages. For example, air temperature at 12:00 will very often be higher than the daily average and presumably ice conditions correlate more with the daily value than the 12:00 value. For wind speed I guess there is a similar problem: Strong winds during nighttime would have an impact on the sea ice conditions (used for training), but the input data for wind cannot reflect this. There is nothing wrong using 12:00 data, I'm just wondering if the model could do better if daily data were used.

C) I like that the text refers to figure numbers more often now. However, please double-check if the correct figure numbers are used throughout the manuscript. I did find some mistakes.

D) 321-323
Why do you replace an unusual break-up date by a freeze-up date? Wouldn't this make the mistake even bigger? Couldn't you instead extent the allowed break-up range to include all possible break-up dates?

E) Why is the comparison to ice charts a subsubsection while the comparison to S2S is a full section? Spontaneously, I would rather suggest to make one section for the evaluation of the spatial full-domain forecasts and another section for the evaluation of forecasts at point locations. In any case, just give the section-layout another thought.

Abstract
###############

If there is space left, I would suggest to mention the most important 1-2 results.

Introduction
################

"have included"
Could also be written in present tense.

"(Drobot et al., 2006),"
Remove comma

"by Zhang et al. (2008) in which"
I would use a comma before "in which"

The sentence "A comparison..." comes quite suddenly - I don't see a connection to the previous statements. Did you delete this introducing sentence on purpose? "The majority of studies on sea ice prediction and forecasting focus on the pan-Arctic domain"

50, 51
Are you aware that you use "spatio-temporal" in the title and in the abstract but "spatiotemporal" otherwise?

Also here: "sequence to sequence" in the title but "sequence-to-sequence" otherwise. Did you choose to do it like this on purpose?

Bushuk
The given DOI number "10.1022/2017/GL073 155" does not work. It should be "10.1002/2017GL073155"

Data
####################

71-72 "When ... ERA5 the SIC..."
"When ... ERA5, the SIC..."

"network.Recalling"
Missing space?

"was set to zero." and 84 "was set to the average"
Consider to use "we set ..." instead of passive past tense to make it more clear what was found in ERA5 and what was done by you.

**3 Study Region**
################

Figure 1
Great to have this map now! If you want to optimize it, you could plot the red dots on top of the grey landmask. In Python and matlab the plotting order can be set with the option "zorder".
Additionally, the titles (port names) of the six subplots could be placed more consistently, e.g. flush-left.

Caption of Figure 1 "model grid point near Quaqtaq"
Looking at the subplots for all ports (Churchill, Inukjuak and Quaqtaq) I don't see how/why Quaqtaq is a special case. All model grid points seem to be located in the water region. (And considering that you are working with a sea ice model, that makes a lot of sense.)

**4 Forecast model architecture**
##############################

98/99 "video captioning (...), speech"
"video captioning (...), and speech"

"the number days"
"the number of days"

'and "Augmented Model"'
' and the(?) "Augmented Model"'

Would be helpful to mention that one only needs to look at the black part of Figure 2a for the Basic model.

Caption of Figure 2
- Add a space in "architecture(a)"
- Not coming from the ML field, I don't know what an "adder" is. But if/as this is not critical for understanding the basic principle of your model, it's ok for me to not go into more detailed explanations here.

**5 Description of Experiments**
#############################

space missing after "operationally,"?

148, 149, 155 etc.
It seems the concept of "model weights" has not been introduced/explained. I also don't see it in Figure 2.

157-159
I find these sentences confusing. Isn't the "For example"-sentence saying the same as the sentence starting with "The model for year 11"? The figure helps a lot to understand what you want to say :-) Would it maybe be easier to use year numbers like 1995 and 1996 instead of 11 and 12?

Figure 3
- "Train" in the legend for the blue color means "Training" I guess.
- I know "testing" is a term in ML, but it could be confusing that you call the green color "Test" while you talk about "prediction" in the text.
- Just a little detail, but it makes me feel unsure:
At the moment the figure indicates that weights from the combination of Train-1995 and Val-1996 are used both for Train-1996 and Val-1997. Can you double-check if this is what you want to say? I get the impression that the Weight-arrow should come or end in a colored box instead of the white box?

"points"
Is this the same as pixel?
Ah, no, it's not. Can you connect the next sentence, which explains N, better? And then use "pixel" in line 175. Suggestion:
"... N is the total number of points considered. For monthly scores, N is the product of the number of pixels in the spatial domain, ..."

Two times "binary accuracy" in this sentence. Maybe change to "... impact the resulting score."

"grid location"
Consider to use "pixel" again?

"points in the spatial domain"
pixel or point?? I thought N was the number of points.

"the average of 'ERA5' sea ice presence"
Why did you drop to mention ERA5? I find it is an important information.

"extracting"
Not sure I understand where you extract from. I thought you would just need to "run" the April-model to get the 90-day forecasts.

"test data in Fig 3"
This is confusing to me: The arrow in Fig 3 indicates that the green Test-data originate from the trained/validated model. I thought this is the forecast output not the ERA5 observation data.

7.1.1 Monthly averaged results
################################

"each value"
"the value", because there is only one value for each (i,j).

"(a) and (b) of Fig 5"
Figure 4? And why not (c), as mentioned in the sentence before? Maybe can be left out?
"For example, the value at index (i,j) of each panel represents..."

200-201 "This continues... April 1-30"
I suggest to remove this sentence and instead show the example for index (2,1).

"Panel 1b)"
Fig 4a?

Suggestion: Start the sentence with "Fig 2 (a,b,c) show that the binary accuracies ..." to help the reader find where we are.

206, 207
If "Basic AND Augmented models" then Fig 4d and 4e.

"early lead times"
"short lead times"?

208-209
Do you talk about Fig 4f now? "Improvements" is ambiguous.

"early lead days"
"short lead times"

Figure 6
It would be nice if the aspect ratio of x and y axis was 1, so that the dashed line would be at 45 degree.
(I have pointed this out last time already and you have marked it as "Modified", but I don't see any modification...?)

7.1.2 Spatial maps of sea ice presence
############################################

I still can't see where in the northern part of the domain the Basic model has higher ice presence probability than the Augmented model.

7.2.1 Freeze-up and Break-up Accuracy
#######################################

"is in incorrect"
"is incorrect"

Section 7.2.1 basically describes a new skill score, and hence it would actually fit well into section 6. Can you move it there to give a full overview about all used metrics in section 6?

7.2.2 Freeze-up and Break-up in comparison with ERA5 data
##################################################

"lead days The"
Add ".".

"Fig 9b"
"Fig 8b"?

Figures 8 and 9
Add a space between the subfigure number and the subfigure caption
(I have pointed this out last time already and you have marked it as "Modified", but I don't see any modification...?)

261-263
This sentence is misleading. It sounds like you compare the pattern of the Basic model with the pattern of the Augmented model. But I assume you are actually comparing both models to the Climate Normal?

Why in past tense?

264, 265 vs. 269, 261, 263, caption of Fig 10, 290
Inconsistency in use of "lead day" and "lead days". (I didn't check the whole document if there are more cases.)

"((c) and (d))"
(Fig 10c,d)

"influx of freshwater inflow"
Couldn't "influx" be removed?

"Quataq"
"Quaqtaq"

"CIS"
Has this abbreviation been introduced already?

Captions of Figures 11 and 12
Is it possible to use a proper "+/-"sign?

302-303 "as for break-up as for freeze-up"
Still not quite... ;-) Now there is one "as" too much.

7.2.3 Ice charts
###################################

Section title: Do you want to make the wording of 7.2.2 and 7.2.3 more similar?

"described section in"
"described in section"

310-311 & 312 "CIS charts"
"ice charts"

"assimilation"
I don't think this term can be used here.
"Ice charts are compiled by ice analysts who manually combine information from different sources like ..."

and elsewhere "ERA-5"
Be consistent: "ERA5" or "ERA-5"

The description of the CIS ice charts and Ice Atlas should actually go to section 2 "Data", shouldn't it?

"2 metrics"
How is this now different to what we have seen in the previous section? Is the "Accuracy" you mention here the same as before?

How exactly do you get a forecast from an Atlas?

Refer to Table 1?

Table 1
Missing full stop at the end of the caption.
Sounds like the selected sites are shown in the Ice Atlas.

Table 2
Explain "MAE" in the caption and give the unit (days).

S2S System
####################################

Should the introduction of the S2S data go to section 2 "Data"? Or at least be mentioned there for people who go to "Data" to look up info about used datasets.

Figure 13
Could you add the month names to the subfigure caption (like "a) May" instead of "a)")?

Caption of figure 13
Move the second full stop from the end to the beginning of the last sentence.

"(false negative, FN)"
Use math font for "FN"

Figure 14
Binary accuracy was defined as (TP + TN )/N, which is a scalar quantity for the whole domain. Also false positive rate and false negative rate are scalar quantities, one number for the whole domain, if I understand correctly. I miss an explanation how you can plot those quantities as a map. Maybe you show probability of false negatives (without "rate")? And something like probability of correct prediction/hit instead of binary accuracy?

Caption of Fig 14 and 15
Needs improvement/more details.

370, caption of Fig 14/15, General reminder
Inconsistent use of "lead day 30", "30 lead days"

370-371 "maps showing ... are shown."
One "show" should be enough.

For people who don't read the paper in a linear way, can you quickly repeat what observations (="truth") you use to calculate the hits, FP and FN.

"July 1 to July 30th"
Either "1st" or "30"

double "are"

"that"
"than"

and 390 and 393 "proposed approach"
Better stick to "Basic and Augmented models". At least use plural form. Or Seq2Seq models?

and elsewhere
When you refer to "portion of domain", do you mean Hudson Bay or the whole model area? In my view, Kivalliq is pretty much in the center of the domain (latitude 61N out of 52N to 70N). [I just realize: Maybe this could also be an explanation for my comment about line 243?]

grammar correct?

Discussion
#########################

"spatial-temporal"
spatio-temporal

393-394
Reference to a figure?

Your manuscript does not have a Figure 1d.

398-402
Quite lengthy text with potential to be shortened.

"enforce"
This word sounds weird to me in this context. Maybe more something like "incorporate/draw on knowledge from the climate normal"? (But I'm not a native English speaker...)

"Fig 13 and 14"
and 15?

"they"
their

Conclusion
#######################

"in advance"
Isn't lead time always in advance?

434-435
Suggestion: "The Basic model uses eight input variables from the ERA5 dataset for the 3 days prior to the forecast start day."

"basic model"
"Basic model"

"where it"
"which"

"climate variables ... over the forecasting period"
You mean Climate Normal? But this is historical data, so then it cannot be used 'over the forecasting period'. Can you clarify the text?

"climate normal"
You improved the consistent use of capitals a lot. However, this paragraph seems to be an exception...

438, 207, 225, 396
I still find "short lead times" better than "early lead days".

"probability assessment by calibration analysis"
I don't quite know what you are trying to refer to.

"lead day"
"lead time" (You also wouldn't say "Air temperature is decreasing with meter." but "Air temperature is decreasing with height.")

"operational"
I guess you are thinking about dynamical forecasting here. A statistical approach like yours could also be "operational".

Paragraph 444-452
The content of this paragraph is pretty good but the sentences are a bit unconnected. Maybe it can be improved by starting the paragraph with the last sentence with a structure something like this: "The model is demonstrated in hindcast mode here, but it is intended to be used for forecasting. We do not envision this to be difficult .... Using different input data requires fine-tuning of the weights, but this is a quick process and only takes around 15 minutes .... Producing a forecast only takes 10 seconds which is much less than with commonly used dynamical forecast systems. This clearly demonstrates the advantage of the Seq2Seq approach. ... " Or so.

"different reanalysis"
A reanalysis product is produced by re-running the atmospheric model (including assimilation) in hindcast mode in a more consistent way than it is possible to do for the "analysis product" (which is used as initial conditions for real forecasts). Hence, reanalysis products are usually only available with a considerable delay (months). Therefore I don't see that you could run your model in forecast mode based on reanalysis data as input.

Paragraph 452-462
Good point with the coarse resolution which limits usability for mariners. My understanding is that dynamical forecast models have difficulties to go to higher resolutions for long predictions because the computational costs are so high. But as computational efficiency seems to be the big advantage of your approach, I wonder: Could your Seq2Seq-model be trained with high-resolution atmosphere/ice data (e.g. from an analysis product of a regional atmosphere-ice-ocean model) and with this be able to produce high-resolution sea ice predictions in a computationally cheap way?

"et al"
"et al."

Code and data availability
####################
What about the ice charts and Ice Atlas? (I don't know if they need to be mentioned if they are not available.)

This link returns "Page not found".
https://cds.climate.copernicus.eu/cdsapp/dataset/reanalysis-era5-single-levels?tab=overview

In general for links I would expect info about when the page has been accessed last.

"Vitart et al."
Only two authors, therefore "Vitart and Robertson"

"available"
"is available"

Reminder to replace the place-holder.

Acknowledgments
######################
I thought Copernicus had some rules about what to write as acknowledgments when one uses data from CDS.
ERA5 was downloaded from CDS, wasn't it?

References
##################

Please check the corrections I've provided in the first review. A few of them are still not implemented.

Missing page numbers and misspelled Danish letter in Dybkjær:
Tonboe, R. T., Eastwood, S., Lavergne, T., Sørensen, A. M., Rathmann, N., Dybkjær, G., Pedersen, L. T., Høyer, J. L., and Kern, S.: The EUMETSAT sea ice concentration climate data record, The Cryosphere, 10, 2275–2290, https://doi.org/10.5194/tc-10-2275-2016, 2016.

Missing volume number and missing initial letter for second author:
Vitart, F., Robertson, A.W. The sub-seasonal to seasonal prediction project (S2S) and the prediction of extreme events. npj Clim Atmos Sci 1, 3 (2018). https://doi.org/10.1038/s41612-018-0013-0

Missing references to Gignac et al. (2019) and Dirkson et al. (2019) (line 414-415)
Also add full stop after "et al" in line 414-415.

---

## Referee Report (RR2)

**"Probabilistic Spatio-temporal Seasonal Sea Ice Presence Forecasting using Sequence-to-Sequence Learning and ERA5 data in the Hudson Bay region"**

The manuscript has improved a lot. The text is more readable and consistent, figures are referenced correctly, and the overall structure seems logical to me now.

I'm listing some more minor points below. In few cases, especially in the reference section, I repeat myself from previous reviews because I think you might have overlooked my comments. The biggest remaining issue in my opinion is the citation/acknowledgment of the various datasets in "Code availability" and/or "Acknowledgment" sections.

Unfortunately, the track-change document you provided is only partly helpful for my task as it sometimes pretends text would have changed even though it didn't (e.g. section 6.3) and sometimes it pretends text would not have changed even though it did (section about Code availability). It would save the reviewer quite some time if one could rely on the track-change document.

################################

21 "(Guemas et al., 2016)"
 Guemas et al. (2016)

Figure 3
I'm happy to hear that you found my comments helpful. But then I'm confused that the figure is exactly the same as before...?

189 "to which are model forecasts are"
remove first "are"

189 "are compared are"
are compared to(?) is

197 "Section 2"
or more precisely: Section 2.2

226 "and if not the prediction"
and if not, the prediction

I don't understand why you sometimes use past tense in this paragraph:
342 "showed"
351 "performed"
355 "showed"

366 "bnary"
binary

368 "it can be seen during May"
"it can be seen that(?) during May". Or maybe just "Fig 13 shows"?

Figure 13
- The letter "a)" under the left panel is almost not readable. Is is partly covered by the plot.
- The legend is a bit small. [Alternatively you could also mention the colors in the text in line 369, like: "... both the Basic and the(?) Augmented models (red) have a higher binary accuracy than the S2S forecasts (black)..."]

Caption of figure 13:
You removed the excessive fullstop, but you didn't add the one between "methods" and "Binary".

374 "(false negative, FN)"
Still the wrong font type for "FN".

377-378
The sentence is OK and correct now. A trick that would make it easier for the reader (in my opinion) would be to bring the word "show" earlier in the sentence so that the reader doesn't need to wait for it until the very last word. For example, "In Fig 14 X, Y and Z are shown." requires more mental capacity than "Fig 14 shows X, Y, Z ..."

371-375 Explanation of FP_rate and FN_rate
Thank you for the clarification. After a while I now got the point that you calculate the FP/FN rate in a temporal sense not in a spatial sense. Therefore you now write "in the set of forecasts" instead of "domain". Good. However, in many other occasions in this paragraph you still use the word "points", which I associate more with space than with time. As you work with daily data, could you maybe replace "point" by "day" to make the time perspective more clear? I mean, isn't the FN_rate at a certain grid point basically telling in which percentage of days the forecast of ice-presence turned out to be wrong? And similarly FP_rate tells the percentage of days in which the forecast of no-ice-presence turned out to be wrong. Right?

391 "Basic and Augmented model"
"Basic and Augmented models(?)"

396 "When ice starts to form this"
"When ice starts to form, this"

406-410 "60 day forecast" and "30 day forecast"
I would prefer "60-day forecast" like in line 406 and 408, but just do it consistently.

430 "the the median"
to the median

430 "could. be"
without fullstop. Actually the whole sentence needs to be revised.

431-433

I don't understand what you try to tell me here. Endpoints? Interior points? I think you either need to explain more, or only focus on the conclusion.

435 HadISST

This dataset has not been mentioned before in the paper. Hence it would at least be important to tell whether it is an observational or a model dataset. [Do I conclude correctly that Dirkson used HadISST2 as reference, like you use ERA5 as reference?]

444 "climate normal"

I noted this last time already. Why not "Climate Normal"?

447 "shows the Augmented model"

"shows that(?) the Augmented model"

447 "capable at"

"capable of"

449 "models have substantial improvement"

"have" sounds a bit weird. Maybe "show"?

438 "for freeze-up than break-up"

"for freeze-up than for(?) break-up"

456 "dependences"

dependencies

463 "spatial resolutions"

I would use singular .

466 "our resolution"

I know what you mean but it sounds funny. The resolution of the authors…

465-467

This sentence appears to be quite long and not too concise. I guess you basically want so say something like:

"Hence, in terms of spatial resolution, the ML approach proposed in this study is not coarser than other commonly used approaches, some of which even target marine transportation."

Code availability:

- What happened to this section?? Earlier here was text about S2S for example.
- Ice charts/Ice atlas?

Acknowledgments:
The given link for ERA5 explains how to cite ERA5 data but it is very unlikely the page from which you have downloaded the data. I think you had most of the information in your "Code availability" section last time, but it has disappeared.

References
###############

Still open from my first review:

488 "shipping" -> "Shipping" (Click on the doi-link, it really is capitalized)
517 "S., R., G.," -> The co-author is called "Graversen R"
517 "high resolution" -> "high-resolution"
517 missing issue/page number [number = {11},pages = {e2020JC016277}]

Still open from my second review:

556 Missing volume number and missing initial letter for second author:
Vitart, F., Robertson, A.W. The sub-seasonal to seasonal prediction project (S2S) and the prediction of extreme events. npj Clim Atmos Sci 1, 3 (2018). https://doi.org/10.1038/s41612-018-0013-0

496 Bushuk
I see that you tried to correct the DOI number but unfortunately you still have got a typo in there. Please try again:
https://doi.org/10.1002/2017GL073155
And copy&paste it into a browser to make sure you get it right this time ;-)

Additional for Vitart: Make sure to also copy the doi-link I give you. The one you use links to a different paper (Vitart et al., 2017) than the one cited here.

---

## Author Response (AR2)

With regards to the editor's comment, we added a section on "Code and data availability" to the revised manuscript.

---

## Author Response (AR3)

**Reply to referee 1**

We would sincerely like to thank the reviewer for the comments. We have made the suggested changes. We have also put the table information directly into the latex file so it no longer shows up with a line underneath.

**Reply to referee 2**

We would sincerely like to thank the reviewer for the very thorough job and excellent comments. We provide responses below for the items that required a response

**General comments**

We have reviewed and revised Section 7.2.3, 8, 9 and 10 to improve readability as suggested by the reviewer.

B) We did not test daily data. On the one hand, the reviewer has made the good point that by only using the noon data we may be missing events happening at different times of the day. However, averaging the states would blur these events (e.g., an opening in the ice cover). On the other hand, by training the model with a relatively large set of noon data, we may have a representative selection of instantaneous events in the training data.

We thank the reviewer for bringing up this point. It relates to the question of time-correlated events, which would be interesting to investigate in a separate study.

C) Thank you for pointing this out. We have double checked the figure numbers

D) We agree extending the break-up date period analysis would have simplified the analysis and improved the final results. The methodology consisted of replacing missing values by the freeze-up date in order to handle multi-year ice situation when no break-up dates are available. We acknowledge the late breakup dates bias from ERA5 is not captured by the break-up analysis (for this area requested by reviewer) and is artificially degrading the results compared to CIS ice charts.

E) The comparison against ice charts is not its own section because the break-up and freeze-up are assessed, using the same method as for the ERA5 data. Hence, it is a subsection of the break-up and freeze-up accuracy assessment. We have also changed the placement of the S2S comparison, putting it inside the results section (section 7).

**Abstract**

**Introduction**

All corrections have been made.

**Data**

All corrections have been made.

**Forecast model architecture**

All corrections have been made.

**Description of Experiments**

All corrections have been made. We also included a brief explanation of model weights. Thanks a lot for the comments about Figure 3. Very helpful!

**Monthly Averaged Results**

All corrections have been made.

**Spatial maps of sea ice presence**

We removed the part about the Basic model having higher ice presence probability in the northern portion of the domain as this is a minor point.

**Freeze-up and Break-up  Accuracy**

All corrections have been made. A new subsection has been made in the section 2 (Data) to describe the ice charts. The accuracy here is different than that shown in the binary accuracy maps because it is calculated by checking if the ice concentration passes a threshold and stays above or below the threshold for a given length of time, as opposed to checking each day. This is now described in section 6.3 (Freeze-up and Break-up Accuracy). We also clarified the wording leading to confusion about a forecast coming from an Ice Atlas.

**S2S system**

All corrections have been made.  A new subsection has been made in Section 2 (Data) for the S2S data. The description of false negative rate and false positive rate has been clarified to explain how it is computed. We erroneously stated that it was calculated over the entire domain. However, these rates were calculated over the given set of forecast data.

**Discussion**

All corrections have been made

**Conclusions**

All corrections have been made. We modified the wording for paragraph lines 444-452, and clarified the wording for the use of a reanalysis. Regarding the question about using high-resolution data, this is something we are currently doing. The Seq2seq model can be trained in that manner, but it is more computationally demanding. Hence there will be trade-offs between resolution, data record length (in years) and domain.

**References**

The reference to Vitart et al. was taken from the Nature website and cited using \citep in latex. Other changes to the references have been made. We thank the reviewer again for their attention to detail.

---

## Author Response (AR4)

Dear Reviewer,

We would sincerely like to thank you for your thorough review and apologize for some shortcomings in the previous round. We have compared old and previous more carefully this time using latexdiff. Please note we were not able to figure out how to show the mark-ups in the references, but you can check that in the revised version they have been corrected. However, we were not able to show the volume number as requested, although this information is in our .bib file. After checking other papers in the Cryosphere, we don't see volume numbers (though of course it was not a complete search), hence we are not sure if this is something in the Copernicus.bst file.

Below we list our response to each comment

21 modified to Guemas et al. (2016)

Figure 3 – this figure has been revised so the arrow goes from the weights box in one iteration to the weights box in the next iteration, which was the piece you pointed out that was very helpful. We also changed the legend to be more consistent with terminology in the text. Although this figure was updated for the previous revision, the path to the figure was not correct and therefore it did not show up in the revised version. Sorry about the confusion.

189 "to which are model forecasts are" – we removed the first "are"

189 "are compared are" – changed to "are compared to"

197 changed to Section 2.2

226 changed to "and if not, the prediction"

342/351/355 We also used past tense in the previous paragraph

366 corrected "bnary" to "binary"

368 changed to "Fig 13 shows"

Figure 13 Moved the (a) and made the legend bigger. Removed the excessive full stop in the caption, and added one between "methods" and "Binary".

374 Changed to a mathfont for FN and FP

377-378 Changed to "Fig 14 shows…."

371-371 – Changed to the use of days instead of points

391 Changed to "Basic and Augmented models"

396 Changed to "When ice starts to form, this"

406-410 Added hyphens so it is "30-day" and "60-day" consistently

430 Changed to "to the median"

430 Revised this sentence

431-433 Removed discussion of endpoints and interior points to focus on conclusion

435 Added information that HadISST2 is an observational dataset. Yes, this was used as a reference dataset in Dirkson.

444 Changed to "Climate Normal"

447 Changed to "shows that the Augmented model"

447 Changed to "capable of"

449  Changed to "models show substantial improvement"

438 Changed to "for freeze-up than for break-up"

456 Changed to "dependencies"

463 Changed to singular

466 Changed to "The spatial resolution used here is similar to that used in studies that carry out seasonal forecasting using a dynamic ice-ocean model (or similar) where a sea ice state vector is predicted as a function of time (Sigmond et al., 2016, Askenov et al. 2017). Hence, in terms of spatial resolution, the ML approach proposed in this study is not coarser than other commonly used approaches, some of which target marine transportation."

Code availability – added information that was there previously about the data availability and changed heading to "Code and data availability" also added ice chart and ice atlas information.

We also obtained a digital object identifier for the code and added this to the "Code and data availability" section.

References: made changes, but volume numbers are still not displaying as noted in the beginning of this reply.